# Structure of PDE3A–SLFN12 complex and structure-based design for a potent apoptosis inducer of tumor cells

Jie Chen [1,5], Nan Liu[2,5], Yinpin Huang[3,5], Yuanxun Wang[1,5], Yuxing Sun[1,5], Qingcui Wu[1], Dianrong Li[1], Shuanhu Gao [4], Hong-Wei Wang [2✉], Niu Huang [1,3✉], Xiangbing Qi [1,3✉] & Xiaodong Wang [1,3✉]

Molecular glues are a class of small molecular drugs that mediate protein-protein interactions, that induce either the degradation or stabilization of target protein. A structurally diverse group of chemicals, including 17-β-estradiol (E2), anagrelide, nauclefine, and DNMDP, induces apoptosis by forming complexes with phosphodiesterase 3A (PDE3A) and Schlafen 12 protein (SLFN12). They do so by binding to the PDE3A enzymatic pocket that allows the compound-bound PDE3A to recruit and stabilize SLFN12, which in turn blocks protein translation, leading to apoptosis. In this work, we report the high-resolution cryo-electron microscopy structure of PDE3A-SLFN12 complexes isolated from cultured HeLa cells pretreated with either anagrelide, or nauclefine, or DNMDP. The PDE3A-SLFN12 complexes exhibit a butterfly-like shape, forming a heterotetramer with these small molecules, which are packed in a shallow pocket in the catalytic domain of PDE3A. The resulting small molecule-modified interface binds to the short helix (E552-I558) of SLFN12 through hydrophobic interactions, thus "gluing" the two proteins together. Based on the complex structure, we designed and synthesized analogs of anagrelide, a known drug used for the treatment of thrombocytosis, to enhance their interactions with SLFN12, and achieved superior efficacy in inducing apoptosis in cultured cells as well as in tumor xenografts.

[1] National Institute of Biological Sciences, 7 Science Park Road, Zhongguancun Life Science Park, Beijing 102206, China. [2] Ministry of Education Key Laboratory of Protein Sciences, Beijing Advanced Innovation Center for Structural Biology, Beijing Frontier Research Center for Biological Structures, School of Life Sciences, Tsinghua University, Beijing 100084, China. [3] Tsinghua Institute of Multidisciplinary Biomedical Research, Tsinghua University, 100084 Beijing, China. [4] Shanghai Engineering Research Center of Molecular Therapeutics and New Drug Development, East China Normal University, 3663N Zhongshan Road, Shanghai 200062, China. [5] These authors contributed equally: Jie Chen, Nan Liu, Yinpin Huang, Yuanxun Wang, Yuxing Sun. ✉email: hongweiwang@tsinghua.edu.cn; huangniu@nibs.ac.cn; qixiangbing@nibs.ac.cn; wangxiaodong@nibs.ac.cn

Molecular glues, small molecule compounds that bring otherwise non-interactive proteins to proximity for protein−protein interactions, have shown promising potential as therapeutical agents, especially for targeting the so-called "undruggable" proteins such as transcription factors and splicing factors[1,2]. In general, molecular glues that influence the activity or fate of their assembled proteins complex, either enhance the gain-of-function effect via protein−protein interaction or induce the target protein degradation by bringing the cellular protein degradation machinery to act on the targeted protein[3].

PDE3A is a phosphodiesterase that catalyzes the hydrolysis of cyclic adenosine monophosphate (cAMP) or cyclic guanosine monophosphate (cGMP)[4]. It appears in three isoforms in cells, each with the distinctive N-terminal regulator domain originating from alternative usage of the start codon, and a common C-terminal catalytic domain[5]. The PDE3 enzymes are involved in the regulation of cardiac and vascular smooth muscle contractility[6]. Several PDE3 enzymatic inhibitors, including cilostazol, milrinone, and anagrelide, have been developed for the treatment of heart failure (cilostazol and milrinone)[7] and thrombocytosis (anagrelide)[8], although the mechanism for the treatment of thrombocytosis has yet to be illustrated.

Schlafen genes are evolutionarily conserved across species with ten members in mice and six members in humans[9]. The biochemical functions of Schlafen proteins are diverse, from inhibition of translation (SLFN12) to prevention of tRNA cleavage induced by reactive oxygen species (ROS) (SLFN2)[10,11]. However, for the majority of the family members, their physiological and biochemical functions are still largely unknown. The signature domain of human SLFN proteins is an AAA ATPase-like domain at their N-terminal regions with a highly conserved "SWADL" (Ser-Trp-Ala-Asp-Leu) domain positioned adjacent to the AAA domain[12]. SLFN11, SLFN13, and SLFN14 have been shown to directly bind RNA[13–15]. Human SLFN14 mutations underlie thrombocytopenia with excessive bleeding and platelet secretion defects but the molecular mechanism for such a defect is not known[16].

In 2016, de Waal and their colleagues found that a synthetic chemical 6-(4-(diethylamino)-3-nitrophenyl)-5-methyl-4,5-dihydropyridazin-3(2H)-one, DNMDP, can induce cell death mediated by PDE3A and SLFN12[17]. Independently, our group later found that high concentration of female sex hormone 17-β-estradiol (E2), also induced apoptosis through PDE3A and SLFN12 and it did so by binding the enzymatic pocket of PDE3A, thus inducing interaction between PDE3A and SLFN12, resulting in the stabilization of the otherwise fast turnover protein SLFN12 in cells[11]. The elevated SLFN12 binds to the ribosome and the signal recognition particles (SRPs), thereby blocking the protein translation on the endoplasmic reticulum, including the translation of anti-apoptotic proteins Bcl-2 and Mcl-1, whose decrease triggers apoptosis[11]. More recently, two more reports showed that two different compounds, anagrelide, and nauclefine, which target PDE3A, also induce apoptosis through this PDE3A and SLFN12 pathway and that the apoptotic activity of DNMDP, E2, nauclefine and anagrelide can all be competitively inhibited by other PDE3A inhibitors, like cilostazol[11,17–19]. These findings together indicate that the promiscuous nature of the PDE3A enzymatic pocket can apparently accommodate small molecules with diverse chemical structures. However, the nature of interactions between those small molecules and PDE3A remains unclear; indeed, it is not known whether these small molecules act to enhance the PDE3A-SLFN12 interaction allosterically (via binding and induction of conformational change(s)) or whether they may function as a "molecular glue" to facilitate PDE3A-SLFN12 complex formation. The question thus calls for the structural analysis of PDE3A-SLFN12 complexes bound to these aforementioned small molecules.

In this work, using Flag- and HA-tagged SLFN12 to pull down PDE3A-SLFN12 complexes in the presence of anagrelide, nauclefine, or DNMDP, we obtained high-resolution cryo-EM structures of those protein complexes. The structures revealed that PDE3A and SLFN12 form a heterotetrameric complex with a butterfly-like shape. Anagrelide, DNMDP, and nauclefine bind to the catalytic pocket of PDE3A with their hydrophilic side and extend their hydrophobic moiety to SLFN12 to form hydrophobic interactions, thus "gluing" the two proteins together. Based on the structure of the complex and details of the binding interactions, we were able to develop anagrelide analogs with enhanced interactions to SLFN12 and thus generated a compound that showed much higher potency in inducing apoptosis in cultured cells and in promoting tumor growth inhibition in tumor xenografts.

## Results

**Anagrelide-mediated cell death depends on PDE3A and SLFN12.** Estrogen-related hormones induce PDE3A-dependent cell death, in a manner independent of PDE3A's enzymatic activity[11]. Our previous screening efforts using an FDA-approved compound library showed that anagrelide dose-dependently induced HeLa cells death (Supplementary Fig. 1a). The cell death induced by anagrelide was abrogated when the endogenous *PDE3A* or *SLFN12* gene was knocked out (Supplementary Fig. 1a). Anagrelide-induced cell death can also be blocked by the co-treatment with PDE3 inhibitors including cilostazol (Cilo) and trequinsin (Treq) (Supplementary Fig. 1b). These findings confirmed the previous studies that both PDE3A and SLFN12 were required for anagrelide to inhibit cancer cell growth[19]. The endogenous SLFN12 protein increased in response to anagrelide treatment. No SLFN12 increase was observed when *PDE3A* gene was knocked out (Fig. 1a).

To investigate the binding details and the cell-killing mechanism of anagrelide, we immuno-precipitated SLFN12 from modified HeLa cells where endogenous *SLFN12* gene had been knocked out and replaced with a cDNA encoding SLFN12 fused to an HA plus 3 × Flag tag. After anagrelide treatment for 12 h, western blotting results showed that (i) anagrelide treatment induced the co-immunoprecipitated of PDE3A with SLFN12 and increased the level of SLFN12 protein (Fig. 1b, lane 2); (ii) co-treatment with the PDE3 inhibitor, Treq, completely blocked the physical interaction between PDE3A and SLFN12, and SLFN12 protein level did not increase (Fig. 1b, lane 3), indicating that the observed increase in SLFN12 protein level is dependent on the anagrelide-induced PDE3A-SLFN12 interaction. Since anagrelide is used clinically to treat thrombocytosis and the drug causes the decline in the overall numbers of megakaryocytes (MKs) in a dose-dependent manner[20], we speculated that anagrelide might induce MK cell number reduction through the PDE3A-SLFN12 apoptotic pathway. To test this hypothesis, we isolated CD34+ cells from fresh umbilical cord blood and cultured them with recombinant human thrombopoietin (TPO) to induce differentiation of these cells into MKs. We then treated the cord blood-derived MK cells with anagrelide alone or co-treated with Treq for 2 days before measuring cell viability. As shown in Fig. 1c, anagrelide induced MK cell death, and the cell death was totally blocked by co-treatment with Treq (Fig. 1c). This observation suggested that anagrelide might function as an anti-thrombosis drug by directly inducing MK cell apoptosis through the PDE3A-SLFN12 pathway, thus reducing the number of platelets, the product of MK cells.

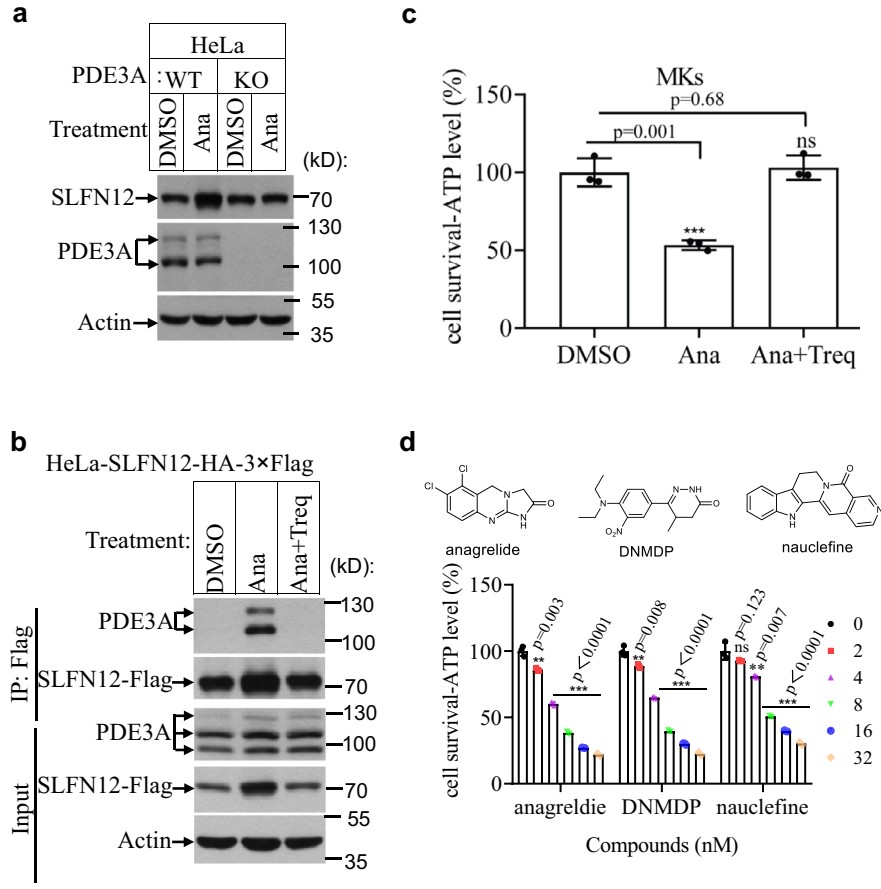

**Fig. 1 Anagrelide-induced cell death required PDE3A and SLFN12. a** HeLa (WT and PDE3A$^{-/-}$) cells were treated with the indicated stimuli for 12 h. The lysates were analyzed by immunoblotting using SLFN12 and PDE3A antibodies. This is a representative result from two independent experiments. **b** HeLa-SLFN12-HA-3×Flag cells were treated with the indicated stimuli for 12 h. SLFN12 was immunoprecipitated using anti-Flag resin. The immunocomplexes and lysates were analyzed by immunoblotting using antibodies as indicated. This is a representative result from two independent experiments. **c** MKs were treated with the indicated stimuli as described in "Materials and methods". Cell viability was determined by measuring ATP levels ($n = 3$, examined in two independent experiments). The data are represented as the mean ± SD of triplicate wells. Student's t-test (two-tailed, unpaired) was performed, ns not significant, ***$p < 0.001$. MK megakaryocyte; Ana anagrelide; Treq trequinsin. **d** HeLa cells were treated with the indicated stimuli at the indicated concentration for 36 h. Cell viability was determined by measuring ATP levels ($n = 3$, examined in three independent experiments). Data are represented as mean ± SD of triplicate wells. Student's t-test (two-tailed, unpaired) was performed, *$p < 0.05$, **$p < 0.01$, ***$p < 0.001$ for comparisons of compounds-treated cells with HeLa cells without compounds treatment (the concentrations of compounds were 0). The chemical structures of anagrelide, DNMDP, and nauclefine are shown.

One of the most interesting features of the PDE3A-SLFN12 complex-dependent apoptosis is that the pathway can be dose-dependently triggered by many structurally diverse compounds (e.g., anagrelide, DNMDP, and nauclefine) and cause cell death in HeLa cells (Fig. 1d). To elucidate the molecular basis of PDE3A-SLFN12 complex formation induced by the aforementioned compounds, we treated HeLa (SLFN12$^{-/-}$) cells with their endogenous *SLFN12* was knocked out that expressed Flag- and HA-tagged SLFN12 (K213R) variant with anagrelide, DNMDP, or nauclefine, and then purified the PDE3A-SLFN12 complexes using Flag-HA tandem antibody pull-down. The SLFN12 (K213R) mutant lost apoptosis-inducing activity but maintained its ability to form a stable protein complex induced by the aforementioned small molecules[11]. We thus chose this K213R mutant since the complex can be accumulated in apoptosis-inducing small molecules treated cells without actually causing apoptosis. The purified complexes, as stained by Coomassie brilliant blue, showed protein bands correlating with that of full-length PDE3A and SLFN12 (Supplementary Fig. 1c), indicating that the structurally diverse chemicals induce apoptosis by forming a complex with PDE3A and SLFN12.

**Cryo-EM structures of SLFN12 and PDE3A complexes formed in the presence of multiple molecular glue compounds**. To investigate the molecular basis of the PDE3A-SLFN12 complex triggered by those compounds, we subjected the purified protein complexes to single-particle cryo-EM analysis and determined the structures of the SLFN12-PDE3A complexes (Supplementary Fig. 2 and Supplementary Table 1) with anagrelide (Fig. 2a; Supplementary Fig. 3a, d; 3.4 Å resolution), DNMDP (Supplementary Figs. 4a and 3b, e; 3.2 Å) or nauclefine (Supplementary Figs. 4b, and 3c, f; 3.2 Å). The main chains of PDE3A and SLFN12 could be unambiguously traced in these structures with most of the side chains of these two proteins nicely assigned, enabling us to successfully build the atomic models. Moreover, the densities of ions were visible in our structures, exemplified by the Zn$^{2+}$ ion in SLFN12's zinc finger (Supplementary Fig. 5) and Mg$^{2+}$ ions in the PED3A catalytic sites (Supplementary Fig. 6).

Not surprisingly, we found that the PDE3A-SLFN12 complexes induced by all three compounds exhibited the same butterfly-like shape, appearing as a hetero-dimer of PDE3A and SLFN12 homo-dimer (Fig. 2a and Supplementary Fig. 3), containing full-length SLFN12 and the C terminal region of PDE3A (i.e., the

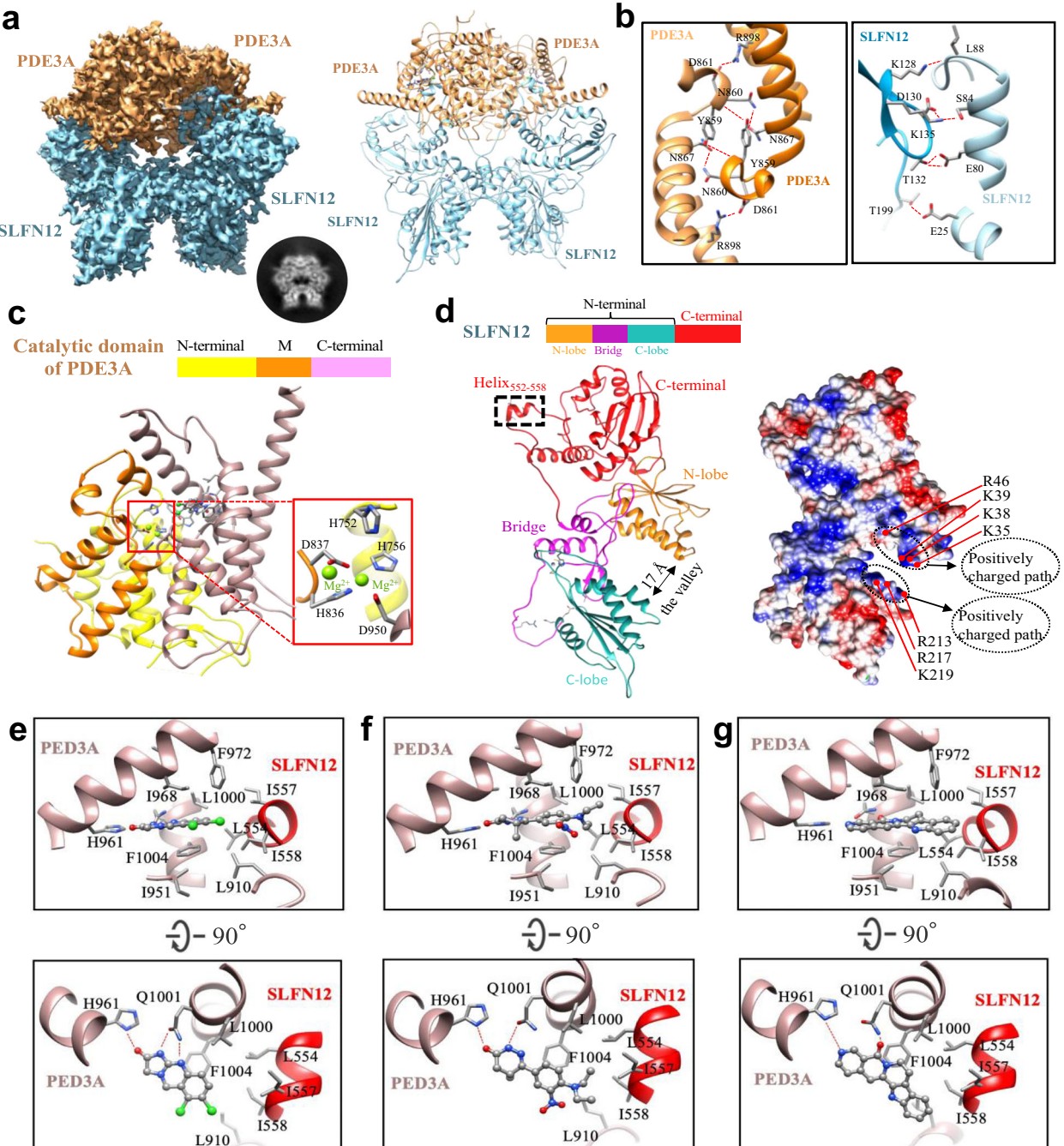

**Fig. 2 Cryo-EM analysis of PDE3A-SLFN12 complexes. a** The cryo-EM reconstruction of anagrelide-induced PDE3A-SLFN12. Left, the cryo-EM density with PDE3A colored in sandy brown and SLFN12 in light blue. Right, the corresponding atomic model. A selected image from the 2D classification was also displayed, demonstrating the butterfly-like shape. **b** The dimeric interaction interfaces of PDE3A (left) and SLFN12 (right). Hydrogen bonds were labeled by red dotted lines. **c** The model of C terminus (K669-Q1102) of PDE3A. The zoom-in image of the $Mg^{2+}$ ions binding site was indicated by the red box. Residue H752 of PDE3A was positioned close to the two $Mg^{2+}$ ions. **d** Left was the model of SLFN12, divided into two domains, the N-terminal and C-terminal domains. The N-terminal domain consisted of two lobes and a bridge motif, forming a U-shaped valley with a diameter of ~17 Å. Right was the electrostatic surface potentials of SLFN12 in the same view as the left model, colored from red (negative) to blue (positive). The positively charged amino acids located at the valley mouth were indicated, forming two positively charged paths. **e** The anagrelide-binding site in PDE3A-SLFN12 complex. Anagrelide was anchored at the binding pocket by forming a hydrogen-bond network with H961 and Q1001 of PDE3A and by hydrophobic interaction with helix$_{552-558}$ of SLFN12. **f** DNMDP-binding site in PDE3A-SLFN12 complex. **g** The nauclefine-binding site in the PDE3A-SLFN12 complex. Hydrogen bonds in (**e**–**g**) were indicated by red dotted lines.

catalytic domain, K669-Q1102). In the PDE3A-SLFN12 complex structures, we identified two hydrogen-bond networks formed at the PDE3A- and SLFN12-homodimer interfaces, located on the opposite sides of the protein interaction interface within the

complex (Fig. 2b). Specifically, at the PDE3A dimeric interface, hydrogen bonds are formed between the side chains of one PDE3A monomer's Y859, N860, and D861 residues with the main-chain oxygen of L858 and the side chains of the N867 and

R898 residues of the second PDE3A monomer (Fig. 2b, left part). For the SLFN12 dimeric interface, hydrogen bonds are formed between one SLFN12 monomer's K128, K135, T132, and T199 residues with the L88, S84, E80, and E25 residues of the second SLFN12 monomer (Fig. 2b, right part). The hydrophobic interaction mediated by anagrelide plays a central role at the PDE3A-SLFN12 heterodimer interaction interface, although there are also three hydrogen bonds dispersed at distant sites (Supplementary Fig. 7). Garvie et al. also solved the structure of PDE3A/SLFN12 complex with DNMDP and reported the similar results recently[21].

PDE3A's C terminal region contains the catalytic domain for hydrolysis of cGMP/cAMP (Fig. 2c)[22]. Our model indicated that this region adopts a similar compact 3D conformation as its isoform PDE3B[23], which is composed of 16 α-helices divided into three subdomains: the N-terminus, M (middle), and C-terminal subdomains (Fig. 2c). The catalytic site at the junction of these three subdomains was visible, comprising the H756, H836, D837, and D950 residues that chelate two metal cations. H752 at the N-terminal subdomain is known to function as a proton donor during cGMP/cAMP hydrolysis and this residue was also positioned nearby the catalytic site in our PDE3A model. The substrate-binding pocket is located at the C-terminal subdomain that anchors anagrelide, DNMDP, and nauclefine mainly through hydrogen-bonding interactions (Fig. 2e−g).

The N-terminus of SLNF12 is composed of two lobes connected by a bridging domain, together generating a U-shaped "valley" with a ~17 Å mouth. Two positively charged patches are located at the entrance of the valley (residues K35, K38, K39, R46, and R213, R217, and K219). Note that the structural information of the C terminal region is absent in previously published structures of SLFN family proteins[14]. Our complex structures indicated that SLFN12's C-terminal region is present as a separate domain, extending away from the N-terminal region via a flexible loop connection (E343-S350). In our structure, a short helix (residues E552-I558) in the C-terminal region of SLFN12 projects out from the main SLFN12 body (Fig. 2d), reaching PDE3A to form a hydrophobic groove, which serves as the heterodimeric interface between SLFN12 and the PDE3A-bound compounds (Fig. 2e−g).

Although anagrelide, DNMDP, and nauclefine are structurally diverse molecules, our structural analysis indicated that all of them occupied the same binding pocket in PDE3A (Fig. 2e−g) and took the similar interaction mode within the PDE3A-SLFN12 complex. The hydrophilic moieties of these molecules containing carbonyl and amine groups anchor inside the PDE3A enzymatic pocket, while their hydrophobic aryl rings or alkyl tails extend outside of the pocket (Fig. 2e−g). Taking anagrelide as an example (Fig. 2e), the hydrogen-bonding interaction networks are formed between PDE3A enzymatic pocket and the hydrophilic imidazolidinone moiety of anagrelide. The H-bond network involves the carbonyl group of anagrelide and the nitrogen in the side chain of H961, and the nitrogen atoms of anagrelide with the side chain of Q1001 (Fig. 2e). Additionally, anagrelide is stacked with the phenyl ring in the side chain of PDE3A's F1004. SLFN12's helix552-558 is positioned at the mouth of the PDE3A enzymatic pocket, and the side chains of its L554, I557, and I558 residues comprise a highly hydrophobic moiety for interactions with the hydrophobic phenyl ring of anagrelide, together with the L910, I951, I968, F972, L1000, and F1004 residues of PDE3A (Fig. 2e). These hydrophobic interactions between these three molecules (anagrelide, DNMDP, and nauclefine) and SLFN12 "glue" the PDE3A and SLFN12 together. The cryo-EM structures of PDE3A-SLFN12 complex with these apoptosis-inducing molecules clearly indicate that these molecules (anagrelide, DNMDP, and nauclefine) function as molecular glues.

**PDE3A's catalytic region binds to SLFN12's C terminal region upon anagrelide treatment**. To define the functional domain of PDE3A in anagrelide-induced cell death, we conducted PDE3A truncation experiments. We transfected HEK293T cells with PDE3A truncation variants, including residues 613-1108 (CD-1), 669-1108 (CD-2), 608-1141 (CD-3), 661-1141 (CD-4) and 679-1141 (CD-5) fused with a myc tag at its C-terminus. These variants were stably expressed in a HeLa (PDE3A$^{-/-}$) cell line, in which their endogenous *PDE3A* gene was knocked out. The results showed that truncation variants of CD-1, CD-2, CD-3, and CD-4 could rescue anagrelide-induced cell death, whereas the truncation variant of CD-5 failed to do so (Fig. 3a). Moreover, PDE3A truncation variants (CD-1 and CD-5) were stably expressed in a HeLa (SLFN12$^{-/-}$) cell line expressing Flag/HA-tagged SLFN12. Immunoprecipitation experiments using antibodies against Flag showed that the PDE3A truncation variant of CD-1, not CD-5 interacts with SLFN12 in the presence of anagrelide (Fig. 3b), indicating that the change of cellular viability should be due to aa 669-679, but not aa 1108-1141, and the C-terminus of PDE3A containing residues 669-1108 is sufficient to bind to SLFN12 in the presence of anagrelide.

According to the cryo-EM structures of anagrelide, DNMDP, and nauclefine with PDE3A-SLFN12 (Fig. 2e−g), multiple small molecules with highly distinct chemical structures all bind PDE3A by interacting with the same sites, including Y751, H961, L1000, Q1001, and F1004. To test the relevance of these sites in anagrelide-induced cell death, Myc-tagged wild-type (WT) PDE3A and the variants bearing amino acid mutants (Y751A, H961A, L1000A, Q1001A, and F1004A) were stably expressed in HeLa (PDE3A$^{-/-}$) cells. All of the variants failed to rescue the cell death in response to anagrelide treatment (Fig. 3c), indicating that these sites are essential for anagrelide to bind to PDE3A and induce apoptosis.

To investigate which region of SLFN12 is required for anagrelide-induced cell death, we expressed the full-length SLFN12 and three truncation variants of SLFN12, including residues 1-550, or 1-560, or 1-570 fused with a Flag tag at its C-terminus in HeLa (SLFN12$^{-/-}$) cells. After anagrelide treatment for 24 h, immunoprecipitation with an antibody against Flag showed that PDE3A was co-precipitated with full-length SLFN12 and truncation variants of SLFN12 containing residues1-560 or 1-570 of SLFN12. However, the SLFN12 truncation containing residues 1-550 lost its interaction with PDE3A (Fig. 3d). These results indicated that the residues 550-560 at SLFN12's C-terminus were required for binding with PDE3A, which is consistent with our structural analysis that the residues comprising the short helix (E552-I558) of SLFN12 are essential for the interaction with PDE3A.

To investigate the functional consequence of these SLFN12 truncations, we engineered several HeLa (SLFN12$^{-/-}$) cell lines in which the full-length SLFN12 and three aforementioned truncation variants are expressed under the control of a doxycycline (Dox)-inducible promoter. Upon Dox treatment, the full-length SLFN12 and truncated variants were expressed, and cell death was observed without anagrelide treatment (Fig. 3e). The result indicated that the apoptosis-induction function of SLFN12 was located at the N-terminus of SLFN12. Methyl blue staining that measures the long-term survival of these cells expressing these truncated SLFN12 also consistently showed that the N-terminus up to amino acid residue 550-560 is sufficient to induce cell death (Supplementary Fig. 8).

**Structure-guided optimization of anagrelide generates more potent apoptosis inducers**. Anagrelide is a clinically-used drug with its pharmacokinetics, pharmacodynamics, and safety profiles

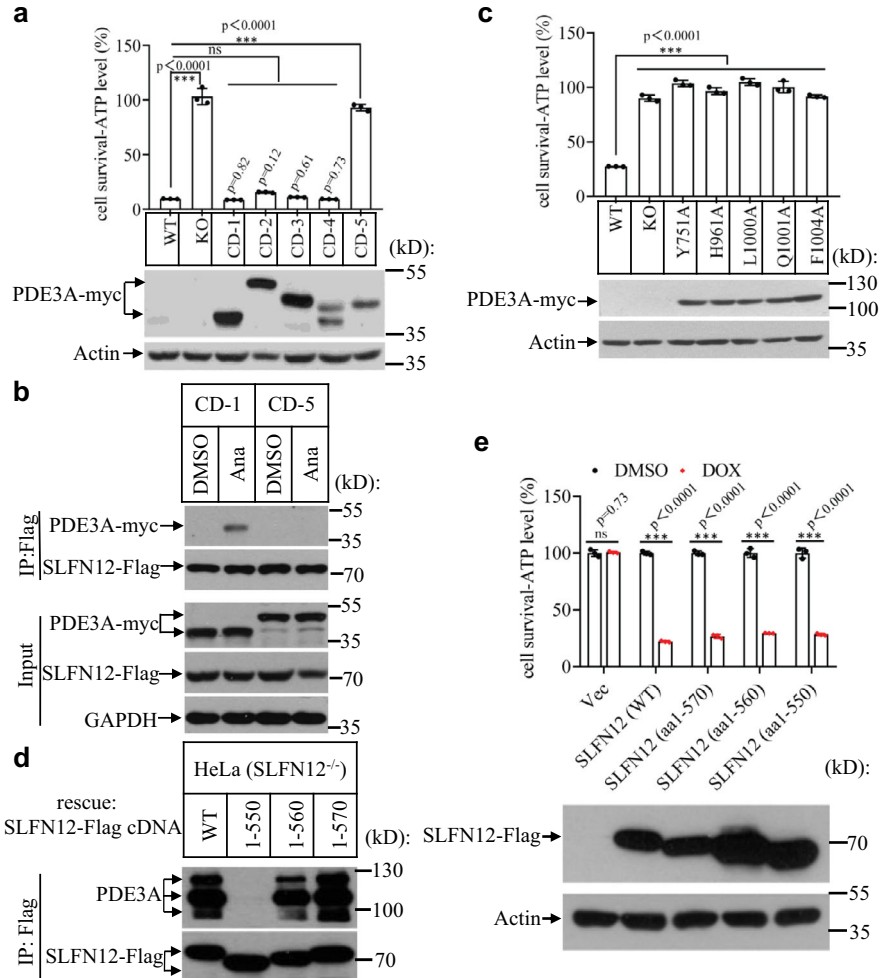

**Fig. 3 PDE3A's catalytic region binds to SLFN12's C terminal region upon anagrelide treatment. a** PDE3A truncation variants including residues 613-1108 (CD-1), 669-1108 (CD-2), 608-1141 (CD-3), 661-1141 (CD-4), and 679-1141 (CD-5) were stably transfected in HeLa (PDE3A$^{-/-}$) cells. The cells were treated with the indicated stimuli for 36 h. Cell viability was determined by measuring ATP levels (upper panel; n = 3, examined in two independent experiments). The data are represented as the mean ± SD of triplicate wells. Student's t-test (two-tailed, unpaired) was performed, ns, not significant, ***p < 0.001. The PDE3A expression was measured by anti-myc immunoblotting. WT Wild type, KO Knockout. **b** PDE3A truncation variants (CD-1 and CD-5) were stably transfected in HeLa (SLFN12$^{-/-}$)-SLFN12-HA-3×Flag cells. The cells were treated with anagrelide for 12 h. SLFN12 was immunoprecipitated using anti-Flag resin. The immunocomplexes and lysates were analyzed by immunoblotting using antibodies as indicated. This is a representative result from three independent experiments. **c** The PDE3A (WT) and different site mutants of PDE3A were stably expressed in HeLa (PDE3A$^{-/-}$) cells. The cells were treated with anagrelide for 36 h. Cell viability was determined by measuring ATP levels (upper panel; n = 3, examined in three independent experiments). The data are represented as the mean ± SD of triplicate wells. Student's t test (two-tailed, unpaired) was performed, ns, not significant, ***p < 0.001. PDE3A expression was measured by anti-myc immunoblotting (lower panel). **d** The full-length SLFN12 and truncation variants including residues1-550, 1-560, and 1-570 fused with a Flag-tag at its C-terminus, were transfected in HeLa (SLFN12$^{-/-}$) cells. After treatment with anagrelide for 12 h, SLFN12 was immunoprecipitated using anti-Flag resin. The immunocomplexes were analyzed by immunoblotting using antibodies as indicated. This is a representative result from three independent experiments. **e** The vehicle, full-length SLFN12, and truncation variants including residues1-550, 1-560, and 1-570 fusion with a Flag tag were inducibly expressed in HeLa (SLFN12$^{-/-}$) cells. The cells were treated with indicated stimuli for 48 h. Cell viability was determined by measuring ATP levels (n = 3, examined in two independent experiments). The data are represented as the mean ± SD of triplicate wells. Student's t-test (two-tailed, unpaired) was performed, ns not significant, ***p < 0.001.

well documented[24]. We thus chose anagrelide over the other two small molecules to further optimization based on the anagrelide-induced PDE3A-SLFN12 structure. The detailed cryo-EM structure clearly indicated that anagrelide acts as a molecular glue to form a stable ternary complex with PDE3A and SLFN12 (Fig. 2e). We noted that anagrelide's acyl guanidine moiety forms hydrogen bonds with the Q1001 and H961 residues of PDE3A. Anagrelide's phenyl moiety engaged in pi−pi stacking with PDE3A's F1004, which was essential for its binding to PDE3. We also observed that the 7-Cl group of anagrelide engages in hydrophobic interactions with PDE3A and SLFN12, specifically with PDE3A's

T844, F972, L910, and SLFN12's I557, I558 (Fig. 2e−g). We then modeled anagrelide analogs with the knowledge that the hydrophobic substitutions of anagrelide at the 7-position are beneficial for the interaction between PDE3A and SLFN12. This notion is also supported by an observation that there was a significant decrease in apoptosis-inducing activity in anagrelide when its hydrophobic 7-Cl group was removed (Supplementary Table 2, compound A14).

We, therefore, speculated that the additional hydrophobic interaction at the 7-position of anagrelide may result in a superior molecular glue. To test this hypothesis, we used the

OpenGrowth[25] program to automatically sample the potential substitutions on the phenyl ring of anagrelide for their effect in promoting interactions between PDE3A and SLFN12. We subsequently used an MM-GB/SA rescoring protocol[26,27] to rank the anagrelide analogs based on calculated binding energy (Fig. 4a). The analogs with potentially enhanced interactions between PDE3A and SLFN12 came from the modification of the 7-position of anagrelide (Supplementary Table 3). Compared to anagrelide, more than one-third of the anagrelide analogs with hydrophobic substitution at the 7-position were predicted to interact more favorably with the residues of SLFN12, especially the analogs with 7-phenyl ring (Supplementary Table 3).

We then synthesized and tested these anagrelide analogs with different hydrophobic substitutions at the 7-position, including aromatic rings or aliphatic chains (Supplementary Table 2). Encouragingly, the apoptosis induction activities of 14 anagrelide analogs increased significantly compared to the original anagrelide, with $IC_{50}$ values ranging from 0.30 to 6.67 nM (Supplementary Table 2). We further modeled the interaction between these anagrelide analogs with PDE3A and SLFN12 based on the structure of the anagrelide-PDE3A-SLFN12 complex (Supplementary Fig. 9). The modeled interactions between the representative anagrelide analogs with PDE3A and SLFN12 (Fig. 4b, f, top panel, anagrelide, compounds A10, A4, A6, and A16) were consistent with our structure-activity relationship (SAR) study. As shown in Fig. 3b, the anagrelide's 7- Cl group engages in hydrophobic interactions with PDE3A's T844, F972, and L910 residues and with SLFN12's I557 and I558 residues (Fig. 4b, top panel). The modeling and the apoptosis-inducing results showed that the hydrophilic pyridyl substitute at this position in compound A10 ($IC_{50} = 6.22$ nM) was not as favorable as the phenyl counterpart in compound A4 ($IC_{50} = 0.56$ nM) in interacting with the hydrophobic interfacial residues (Fig. 4c, d). Further SAR optimization along this direction was conducted and an analog with a p-tolyl substitution (Fig. 4g, compound A6, $IC_{50} = 0.30$ nM) conferred the most potent apoptosis-inducing activity with a ~22-fold improvement over anagrelide ($IC_{50} = 6.67$ nM) (Fig. 4b, e, bottom panel). The fact that higher apoptosis-inducing activity of compound A6 with additional hydrophobic methyl group of the p-tolyl was observed compared to that of compound A4 ($IC_{50} = 0.56$ nM) with the phenyl ring (Fig. 4d, e, bottom panel; Fig. 4g, compound A6 Vs. compound A4) further confirmed that the hydrophobic interaction at 7-position was the key to achieve better molecular glues. In contrast, the analog with a highly hydrophilic carboxylic acid (compound A16) abolished the interaction with SLFN12 (Fig. 4f, top panel) and was unable to induce apoptosis (Fig. 4f, bottom panel).

**Compound A6 shows better tumor growth inhibition than anagrelide**. To evaluate the activity of compound A6 in vivo, we examined its effects on tumor growth in a mouse xenograft model. We firstly inoculated HeLa cells subcutaneously into female nude mice. The tumor-bearing mice were randomized into four groups (n = 5/group) and received the following treatment via an oral gavage daily with methylcellulose: (a) vehicle; (b) anagrelide at 5 mg/kg; (c) compound A6 at 5 mg/kg; and (d) compound A6 at 1 mg/kg. The compound A6 (5 mg/kg) treatment group showed the most dramatic tumor growth inhibition (Fig. 5a–c). For the anagrelide (5 mg/kg) and compound A6 (1 mg/kg) treatment groups, the tumor volume still increased with oral gavage once a day for six consecutive days. We subsequently treated the mice with oral gavage twice a day and the tumor volumes of anagrelide (5 mg/kg) or compound A6 (1 mg/kg) groups dramatically decreased compared with the untreated

group. Notably, no bodyweight reduction was observed in all treatment groups (Fig. 5d).

**Discussion**

Multiple small molecules with highly distinct chemical structures can promote the PDE3A-SLFN12 interaction and induce cell death (e.g., estradiol, DNMDP, anagrelide, nauclefine). Our current structural analysis showed that the catalytic region in PDE3A's C-terminal domain (residues 669-1108) possesses a binding pocket that can accommodate all these molecules. The binding of these small molecules allows PDE3A to recruit SLFN12 and initiate the apoptosis pathway. Our truncation experiments revealed that the patch of residues 669-679 of PDE3A is responsible for the PDE3A sensitivity to these small molecules treatment (Fig. 3a), which is actually located rather far away from the catalytic and enzymatic site, i.e., the molecule-binding site. This probably suggests some allosteric interactions of the N-terminal domain that are required for the proper functioning of PDE3A.

Both SLFN12- and PDE3A-family proteins have been reported to exist as homodimers[23,28,29]. We now found that the dimer of PDE3A-SLFN12 forms a heterodimer structure via hydrogen-bonding networks, which are formed at the dimeric interaction interfaces, supporting the hypothesis that SLFN12 might function as a dimer in vivo. Notably, PDE3A in this study was from the endogenous origin and appeared as its full length in our purification (Supplementary Fig. 1c). Thus, the absence of N terminal density of PDE3A in our cryo-EM structures was likely due to the flexibility of the N terminus, which was averaged out during data processing.

The structure of the N-terminus of SLNF12, with two positively charged patches at the large (~17 Å) entrance of the U-shaped "valley", suggests that this might be the place where it interacts with its downstream effector ribosomes. The single amino acid substitution of K213R that inactivates its apoptotic function is associated in this region.

Although anagrelide, DNMDP, and nauclefine are structurally diverse molecules, our structures indicate that they occupy the same binding pocket in PDE3A (Fig. 2e−g), and all have a similar interaction mode with the PDE3A-SLFN12 complex: their hydrophilic heads containing carbonyl and amine groups are anchored inside while their hydrophobic aryl ring or alkyl tails extend outside of the PDE3A enzymatic pocket to mediate the interaction with SLFN12 (Fig. 2e−g). These hydrophobic interactions illustrate how these chemicals function as molecular glues in promoting the association of PDE3A and SLFN12.

Importantly, based on the sequence analysis and structure comparison with its isoform, PDE3B[23], we found that anagrelide, DNMDP, and nauclefine shared the same binding pocket of cyclic nucleotides in PDE3A. Therefore, the occupation by anagrelide, DNMDP, and nauclefine of the enzymatic pocket of PDE3A would in theory preclude the binding of cyclic nucleotides, thus inhibiting its hydrolysis activity. However, our previous study on nauclefine indicated that such a molecule could also be readily competed out by the cyclic nucleotide substrates of PDE3A, making it appear that nauclefine was unable to inhibit the PDE3A enzyme activity when sufficient cAMP/cGMP was present in the reaction[18]. The nature of the molecular glue of anagrelide, DNMDP, and nauclefine is clear from their complex structures with PDE3A and SLFN12. The interactions require the hydrophilic moiety of the molecule to bind to the enzymatic pocket of PDE3A while the hydrophobic moiety of the molecule extends out to form an interaction with SLFN12, which would not be able to form stable interactions with PDE3A without the contribution from these small molecules. The structural information also

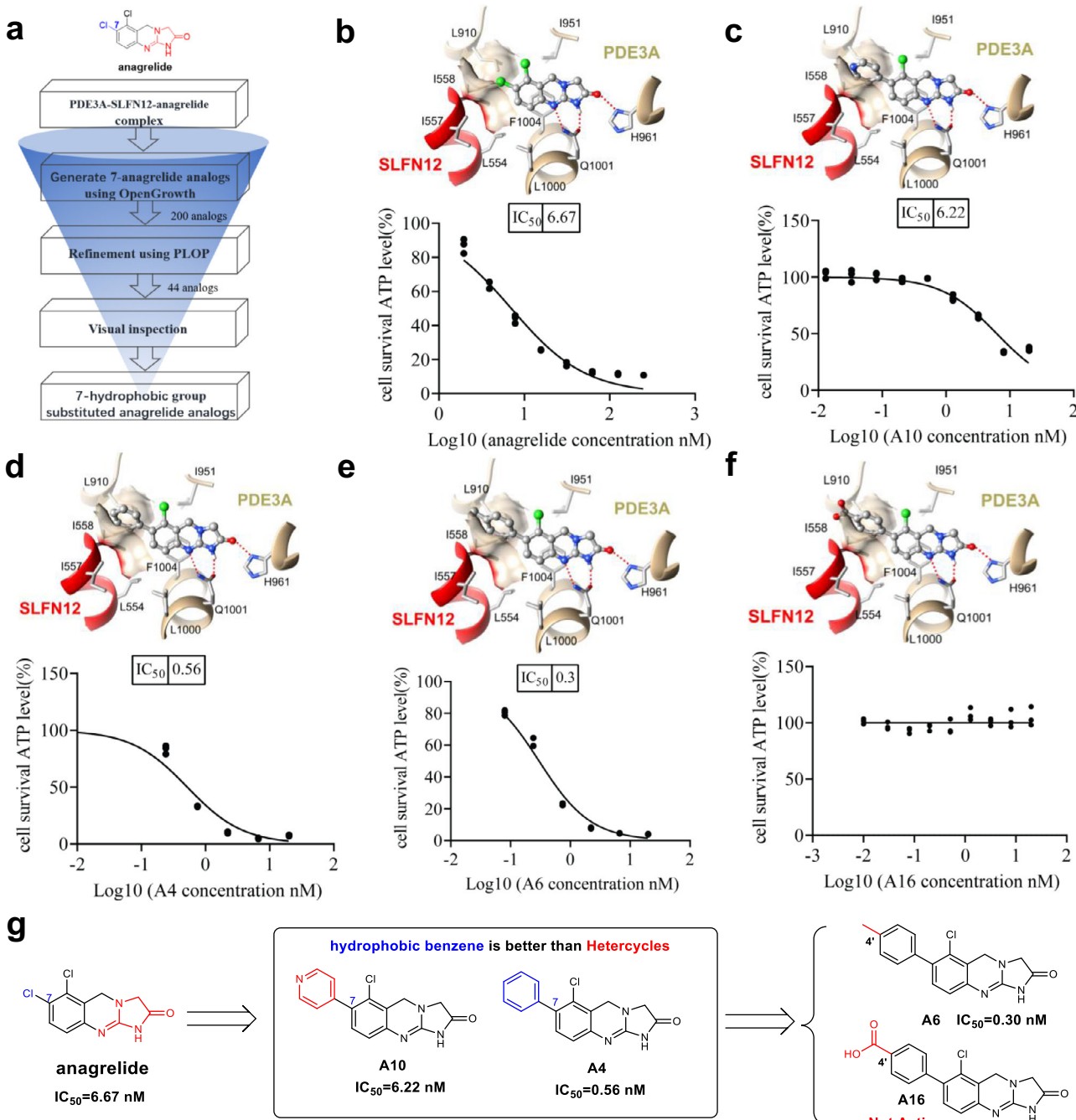

**Fig. 4 Structure-based design of molecular glue. a** Flowchart of our structure-based design strategy of anagrelide analogs. **b** Focused view of the binding mode of anagrelide, where the favorable hydrophobic interactions are formed between the 7-chlorine atom in anagrelide and PDE3A-SLFN12 interfacial residues (L910, L554, I557, and I558). **c** Docking mode of compound A10, with the hydrophilic pyridyl substitute interacting unfavorably with the hydrophobic interfacial residues. **d** Docking mode of compound A4, with favorable hydrophobic interactions forming between the phenyl substitute with the interfacial residues. **e** Docking mode of compound A6, the *p*-tolyl group forms the most favorable hydrophobic contacts with the interfacial residues. **f** Modeled binding mode of compound A16, the negatively charged carboxyl group is not complementary with the hydrophobic interfacial residues. In **b–f**, SLFN12 and PDE3A were shown as red and sandy brown ribbons, respectively. The anagrelide and its analogs were represented by the ball stick. The red dashed line indicated hydrogen bonds. HeLa cells were treated with the indicated stimuli for 36 h, cell viability was determined by measuring ATP levels (n = 3, examined in three independent experiments), and IC$_{50}$ values were calculated using GraphPad Prism. The data are represented as the mean ± SD of triplicate wells. **g** Summary of the structure-activity relationship of the representative anagrelide analogs.

provides clear guidance on how to make inhibitors of this apoptosis pathway, i.e., deleting the hydrophobic moiety while enhancing the hydrophilic interactions; and to make more potent apoptosis inducers by increasing the hydrophobic interactions between the small molecules and the I557 and I558 residues of

SLFN12. We also analyzed the correlation of the logP value and IC$_{50}$ of the molecules with the similar interaction pattern at this interface and indeed observed the same tendency that the greater the value of LogP, the lower IC$_{50}$ value (Supplementary Note 1). Indeed, we were able to make a more potent compound (A6) that

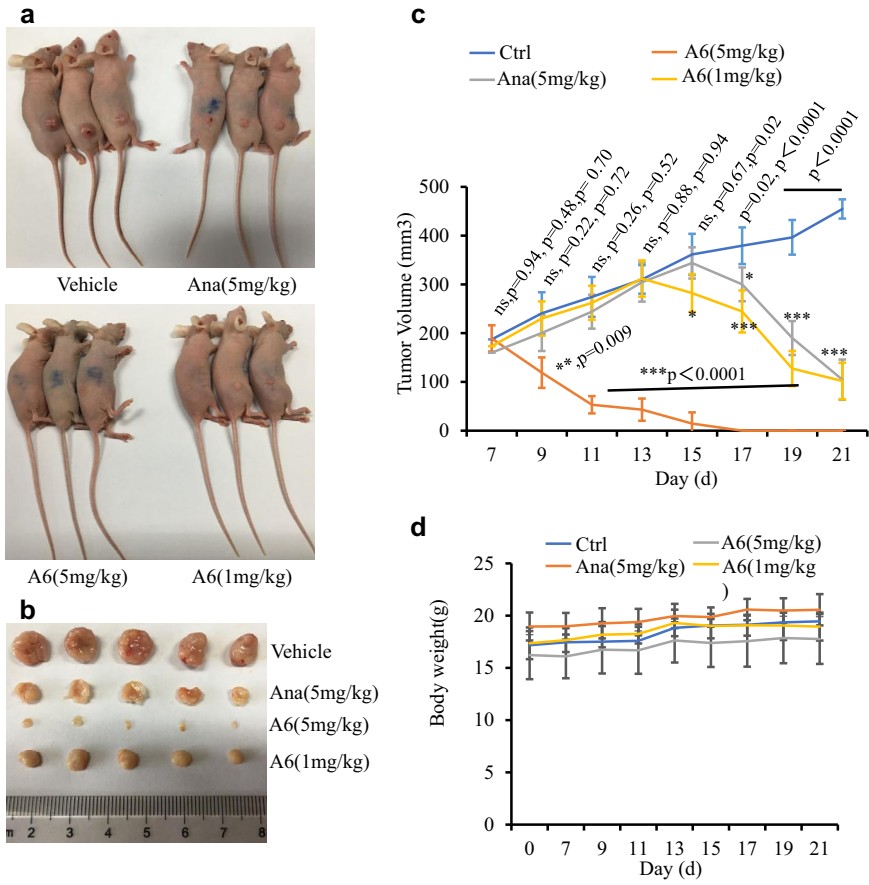

**Fig. 5 Compound A6 shows tumor growth inhibitory effect. a** Female BALB/c-Nude mice subcutaneously inoculated with HeLa cells. After 7 days, the mice were treated with Vehicle, Ana (5 mg/kg), A6 (5 mg/kg), or A6 (1 mg/kg) by oral gavage. Shown are representative photos of mice on day 21. **b** Typical photos of tumors from the Vehicle, Ana (5 mg/kg), A6 (5 mg/kg), or A6 (1 mg/kg) treated mice. Mice were sacrificed after treatment for 21 days; tumors were dissected and photographed. **c** The tumor growth curves of the tumor-burdened mice. Tumor measurements were performed three times a week using calipers; the average tumor volume ± SD for each cohort is displayed (n = 5/group). Student's t-test (two-tailed, unpaired) was performed, ns not significant, *p < 0.05, **p < 0.01, ***p < 0.001 for comparisons of A6 (5 mg/kg)-, anagrelide (5 mg/kg)-, or A6 (1 mg/kg)-treated group with untreated group (vehicle). **d** Body weight curves of the tumor-burdened mice. The average body weight ± SD for each cohort is displayed (n = 5/group).

demonstrates superior cell-killing activity both in cultured cells and in a tumor xenograft model.

Our knowledge of the physiological functions of the PDE3A and SLFN12 pathway has just begun to emerge: from tumor suppressor to placental implant[11,19]. Better compounds based on the structures described in this study will facilitate the study of this critical pathway.

## Methods

**Cells and plasmids.** HeLa and HEK293T cells were obtained from ATCC. psPAX2, pMD2.G, pLVX-Tetone, and pWPI-Cas9 constructs were kept in our lab. Full-length cDNAs for human PDE3A and SLFN12 were PCR-amplified from HeLa cDNA using KOD polymerase (TOYOBO) and were subcloned into pGEM-T vector. Using Quickchange Site-Directed Mutagenesis Kit to generate PDE3A-(mutants) and SLFN12 (mutants) constructs. Myc-tagged full-length PDE3A, HA-3×Flag-tagged SLFN12, and truncation variants were constructed in pWPI. HA-3×Flag-tagged full-length SLFN12 and truncation variants were constructed in pLVX-Tetone vector. Primers were showed in Supplementary Table 4.

**Cell cultures and treatments with test compounds.** All mammalian cells were cultured at 37 °C with 5% $CO_2$. HeLa and HEK293T cells were cultured in DMEM medium (Gibco) supplemented with 10% FBS (Invitrogen, Carlsbad, CA) and penicillin/streptomycin (Invitrogen). Freshly isolated CD34+ cells were cultured with 50 ng/mL recombinant human thrombopoietin (TPO) in StemSpan™ SFEM medium (# 09650, Stem Cell Technologies). After 7 days of culture, cells were counted and then diluted to $1.5 \times 10^5$ cells/mL by addition of fresh Stemspan™ SFEM medium supplemented with 10 ng/mL TPO. Following dilution, cell aliquots of 100 μl were replated into flat-bottom 96-well plates (#3903, Corning) for the cell

survival analysis. At 5 days after the seeding, compounds (anagrelide, 300 nM; trequinsin, 300 nM) or vehicles were added at the indicated concentration for the additional 2 days.

**Coomassie brilliant blue Staining.** Staining solution: Dissolve 2.5 g of coomassie brilliant blue in 450 mL methanol and stir overnight. Then add 450 mL of deionized water and 100 mL of glacial acetic acid. Polyacrylamide gels were stained in the staining solution for 3 h. Then the gels were de-stained in the de-staining solution containing 45 mL methanol, 10 mL glacial acid, and 45 mL deionized water.

**Cell viability assay.** Cell viability assays were performed by measuring the cellular ATP level using a Cell Titer-Glo Luminescent Cell Viability Assay kit (Promega) according to the manufacturer's instructions. The luminescence intensity was read by a microplate reader (Tecan GENios).

**Western blotting.** Cell pellets were collected and re-suspended in lysis buffer (100mMTris-HCl, pH 7.4, 150 mM NaCl, 0.5% Triton X-100, 2 mM EDTA, Roche complete protease inhibitor set, and Sigma phosphatase inhibitor set), incubated on ice for 30 min, and centrifuged at 20,000 × g for 30 min. The supernatants were collected for western blotting analysis. The antibodies used in this research were: PDE3A antibody from Bethyl Laboratories (1:1000, Cat# A302-740A); SLFN12 antibody from Abcam (1:1000, Cat#ab234418); anti-Rabbit-HRP antibody from Sigma-Aldrich (1:5000, Cat# A0545); anti Mouse-HRP antibody from Sigma-Aldrich (1:5000, Cat# A9044); Myc-HRP antibody from MBL (1:10,000, M-047-7);Flag-HRP antibody from Sigma-Aldrich (1:10,000, Cat#A8592); Actin-HRP antibody from MBL (1:50,000, Cat# PM053-7) and anti-GAPDH-HRP antibody from MBL (1:50,000, Cat# M171-1).

**Transfections and virus packaging**. HEK293T Cells were transfected with plasmids using Lipofectamine 3000 (Thermo Fisher) following the manufacturer's instructions. Genes inserted into lentiviral-based vectors were co-expressed with psPAX2 and pMD2.G in HEK293T cells. Eight hours after transfection, the media were changed to high-serum DMEM (20% FBS with 25 mM HEPES). Another 40 h later, the media were collected and centrifuged at $3,000 \times g$ for 10 min. The supernatant was filtered through a 0.22 mm membrane and used to infect cells with polybrene (5 μg/ml). Positively infected cells expressed GFP and were sorted by FACS to establish cell lines stably expressing specific genes.

**Methylene blue staining**. Cells were seeded in six-well plates. After treatment, cells were washed with PBS and stained with 1.5% methylene blue dissolved in 50% ethanol for 10 min at room temperature. Cells were then washed with PBS and photos were taken.

**Flag-HA tandem pull-down**. The cells were cultured on 15 cm dishes and grown to confluence. Cells at 90% confluence were washed once with PBS and harvested by scraping and centrifugation at $800 \times g$ for 5 min. The harvested cells were washed with PBS and lysed for 30 min on ice in the lysis buffer. Cell lysates were then spun down at $12,000 \times g$ for 20 min. The soluble fraction was collected, and the protein concentration was determined by the Bradford assay. Next, 1 mg of extracted protein in lysis buffer was immunoprecipitated overnight with anti-Flag or anti-HA affinity gel (Sigma-Aldrich) at 4 °C. The immuno-precipitates were washed three times with lysis buffer. The beads were then eluted with 0.5 mg/ml of the corresponding antigenic peptide for 4 h or directly boiled in 1% SDS loading buffer.

**Cryo-EM specimen preparation**. A drop of 3.5 μl solution containing SLFN12-PDE3A complex with compounds was pipetted onto EM grid (Au 300mesh, R1.2/1.3) coated by homemade graphene film[30], which had been previously glow-discharged in a low-energy plasma cleaner (Harrick PDC-32G) for 12 s to be hydrophilic. After loading the sample, the grid was transferred into an FEI Vitrobot (Thermo Fisher Scientific) with the humidity of 100% and the temperature of 8 °C and blotted by filter papers (TED PELLA) for 2 s with −2 force. Afterward, the grid was plunge-frozen into liquid ethane and stored at liquid nitrogen.

**Cryo-EM data collection and processing**. Cryo-EM datasets were collected on a Titan Krios (Thermo Fisher) with an accelerating voltage of 300 kV, equipped with a CS corrector and a Gatan K3 summit detector. The movies were automatically acquired by the AutoEMation software written by Dr. Jianlin Lei at Tsinghua University with a total dose of 50 e−/Å$^2$, every of which contained dose-dependently fractionated 32 frames. We collected 2,526 micrographs for anagrelide-induced complex, 2,128 micrographs for DNMDP-induced complex, and 1,395 micrographs for nauclefine-induced complex, respectively. For data processing, we firstly applied MotionCor2[31] to correct the beam-induced motion of individual frames and used Relion3.1[32] to perform the following 3D reconstruction. The CTF values of these motion-corrected micrographs were determined by the CTFFIND4 algorithm[33]. After that, particles were auto-picked and extracted in Relion3.1, which were then pooled into a 2D classification for selecting good particles. After 3D classification and refinement with C2 symmetry, we obtained the structures of Anagrelide- DNMDP- and nauclefine-induced PDE3A-SLFN12 complex at 3.4, 3.2, and 3.2 Å resolution, respectively, estimated by the Fourier Shell Correction (FSC) = 0.143 cutoff criteria. Particle numbers used for the final reconstructions were 82,930 for anagrelide-induced complex, 203,914 for DNMDP-induced complex, and 60,683 for the nauclefine-induced complex.

**Model building**. We built the atomic model of PDE3A based on a previously reported model of PDE3B[23] and for the N-terminal region of SLFN12 based on a reported model of SLFN13[14]. A C-terminal model of SLFN12 was generated by SWISS-MODEL[34], with manual adjustment in COOT[35]. The complex models were finally refined in Phenix[36]. We used UCSF Chimera[37] to analyze the structural files and generate figures in the manuscript.

**Computational details of molecular glue design**. OpenGrowth v1.0.1 was used to automatically generate Anagrelide analogs at position 7[25]. The option of GROWTH_MODE in the parameter file was set to FOG, the list of fragments of Fragments-OpenGrowth-413 was used during the growth. The option of MAX_FRAGMENTS in the parameter file was set to 3. Totally 200 newly generated poses (a total of 44 unique ligands) from the OpenGrowth program were submitted to MM-GB/SA rescoring using Protein Local Optimization Program[38] the details of rescoring can be found in previously published paper[26,27].

**Chemistry**. All reactions were carried out under an atmosphere of nitrogen in flame-dried glassware with magnetic stirring unless otherwise indicated. Commercially obtained reagents were used as received. Solvents were dried by passage through an activated alumina column under argon. Liquids and solutions were transferred via syringe. All reactions were monitored by thin-layer chromatography with E. Merck silica gel 60 F254 pre-coated plates (0.25 mm). 1H and 13C NMR

spectra were recorded on Varian Inova-400 or 500 spectrometers. Data for 1H NMR spectra are reported relative to CDCl3 (7.26 ppm), CD3OD (3.31 ppm), or DMSO-d6 (2.50 ppm) as an internal standard and are reported as follows: chemical shift (δppm), multiplicity (s = singlet, d = doublet, t = triplet, q = quartet, sept = septet, m = multiplet, br = broad), coupling constant J (Hz), and integration. Data for 13C NMR spectra are reported relative to CDCl3 (77.23 ppm), CD3OD (49.00 ppm), or DMSO-d6 (39.52 ppm) as an internal standard and are reported in terms of chemical shift (δ ppm). Samples preparation and purity analysis were conducted on Waters HPLC (Column: XBridge C18, 5 μm, 19 × 150 mm) with 2998PDA and 3100MS detectors, and Waters UPLC (Column: BEH C18, 1.7 μm, 2.1 × 50 mm) with PDA and SQD MS detectors, using ESI as ionization. HRMS data were obtained on a Thermo Q Exactive.

All new compounds were synthesized as indicated in detail in schemes in the supplementary information. Compound A2 was synthesized according to the same synthetic route as compound A1 using different starting materials 3-fluoro-2-chlorobenzaldehyde. Di-substituted compounds A3, A4, A6-A14, A16, A18-A22 were synthesized using Suzuki coupling reaction from A1. Compounds A5 were synthesized according to the same synthetic route as compound A17 using different starting materials 3,4,5-trichloroaniline. Tri-substituted compound A15 was synthesized using Suzuki coupling reaction from A17. A14 was the by-product of the Suzuki coupling reaction for di-substituted compounds. The synthetic details and characterization of these compounds were provided in the Supplementary Methods.

**Animals and mouse xenografts**. HeLa cells ($5 \times 10^6$ per mouse) were subcutaneously injected into female BALB/c athymic (nu/nu) nude mice (age, 4 weeks; weight, 18−20 g). Mice were maintained in an animal facility with 12 h light/12 h dark cycles, temperature (22–24 °C), humidity (40–60%) at the National Institute of Biological Sciences, Beijing. After 7 days, vehicle, Ana (5 mg/kg), A6 (5 mg/kg) or A6 (1 mg/kg) were intragastric administrated once per day for 8 days. Then vehicle, Ana (5 mg/kg), A6 (5 mg/kg) or A6 (1 mg/kg) were intragastric administrated twice per day for 6 days. Tumor volumes were calculated as (length × width$^2$)/2.

**Ethics**. Animal experimentation: Animal care and use followed the institutional guidelines of the National Institute of Biological Sciences (NIBS), Beijing (Approval ID: NIBSLuoM15C), and the Regulations for the Administration of Affairs Concerning Experimental Animals of China.

**Reporting summary**. Further information on research design is available in the Nature Research Reporting Summary linked to this article.

## Data availability

Data supporting our findings in this manuscript are available from the corresponding authors upon reasonable request. Raw blots corresponding to the SDS-PAGE gels, cell viability, chemistry synthesis, and tumor xenograft are included as Source Data file. The cryo-EM maps of anagrelide-, DNMDP- and nauclefine-induced PDE3A-SLFN12 complexes have been deposited in the EMDB under accession number EMD-31103, EMD-31104, and EMD-31105 respectively, and the coordinates in the PDB with PDB ID: 7EG0 [https://doi.org/10.2210/pdb7EG0/pdb], 7EG1 [https://doi.org/10.2210/pdb7EG1/pdb] and 7EG4 [https://doi.org/10.2210/pdb7EG4/pdb], respectively. Source data are provided with this paper and also have been deposited in the Figshare, ID: 14721444. Source data are provided with this paper.

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

## Acknowledgements

We thank Dr. John Snyder and Dr. Alex Wang for critically reading and editing the manuscript. We thank Dr. Jianlin Lei, Dr. Xiaomin Li, Dr. Fan Yang, Xiaofeng Hu, and Cuixia Hu at the cryo-EM and High-Performance Computation platforms of Tsinghua University Branch of the National Protein Science Facility, for the technical support in cryo-EM data collection and analysis. This work was supported by an institutional grant from the Chinese Ministry of Science and Technology and by the Beijing Municipal Commission of Science and Technology (Z201100005320012 to N.H.). This work was also partially supported by the Chinese Ministry of Science and Technology 973 grant (2014CB849603 to X.Q.). We gratefully acknowledge the Beijing Municipal Government and Tsinghua University for their financial support.

## Author contributions

J.C., N.L., and Y.H. were the key contributor in designing and conducting the majority of the experiments. X.W., X.Q., J.C., and D.L. conceived and directed the project. X.W., X.Q., J.C., and N.L. wrote the manuscript. N.L. and H.W. performed the cryo-EM reconstruction and model building of the PDE3A-SLFN12 complex. X.Q., Y.S., and Q.W. designed and synthesized the compound (A1-A22). S.G. synthesized the compound nauclefine. N.H. and Y.W. performed the computational design and molecule scoring based on the complex structure of PDE3A/SLFN12.

## Competing interests

The authors, Xiaodong Wang, Xiangbing Qi, Niu Huang, Hong-wei Wang, Jie Chen, Nan Liu, Yinpin Huang, Yuanxun Wang, Yuxing Sun, Qingcui Wu, and Dianrong Li have filed a provisional patent application (PCT/CN2021/082486) for the application of "Anagrelide analogs and the use thereof". All authors declare no competing interests.
