## [Peer Review File · Nature Communications]

Structure of PDE3A–SLFN12 Complex and Structure-based Design for A Potent Apoptosis Inducer of Tumor CellsREVIEWER COMMENTS

Reviewer #1 (Remarks to the Author):

Chen et al report cryo-EM maps of Anagrelide (and DNMDP), as a “molecular glue” that enhances the interaction between SLFN12 and PDE3A. Structure-based design of analogs led to the development of a 20-fold more potent compound (A6). Anagrelide is a drug used for the treatment of thrombocytosis, and this study further supports the previously suggested mechanism of action of this drug.

Overall, this reviewer is not convinced about the novelty of this work. It is not clear how this work advances the field. Importantly, the major point number 1 mentioned below brings an important concern that needs to be clarified.

MAJOR POINTS

1. The authors claim at the very beginning of the “Results” section: “Our screening efforts using an FDA-approved compound library against PDE3A showed that Anagrelide induces cell death in a dose-dependent manner.” As it is written, it implies that this screen was done now, as part of this paper, but no data is supporting it.

Said screening was performed in a previous paper by the same authors that it is actually referenced in other sections: An et al. 2019. If there was not a new screening, the content and novelty of the paper are significantly lower than if this would have been the case.

2. The authors use the “molecular glue” concept a lot, including in the abstract. However, there is not a single article/review referenced neither in the introduction nor in the discussion. A proper introduction about molecular glues needs to be done.

3. Why the authors decided to focus on Anagrelide (and also a bit in DNMDP) over Nauclefine is not explained.

4. The authors claim in the introduction: “R. An. et al. (2019) showed that a PDE3A inhibitor (Anagrelide) induces interaction between PDE3A and SLFN12 and activates death signaling (An et al., 2019)”.

In the cited paper, it is instead stated:

“... Although we were unable to co-immunoprecipitated PDE3A and SLFN12 after ANA treatment, we did find that both PDE3A and SLFN12 were required...”

This points towards my comment raised just above, why did they focus on Anagrelide?

5. These two claims are too speculative. If the authors want to include these statements, they should include data to show it:

- Speculation about steric effect preventing SLFN12 ubiquitin-mediated degradation
- In the discussion: the occupation by Anagrelide/DNMP of PDE3A would preclude the binding of cyclic nucleotides, thus inhibiting its hydrolysis activity

6. To prove the molecular glue mechanism, it would be nice to show drug-induced proximity between PDE3A and SLFN12 using additional orthogonal strategies (split luciferase, alphaLISA, tethered drug analogs, biotin ligase, FRET...)

MINOR POINTS

1. Figure 4D-G: IC50s should be reported in the figure/legend/text.

2. "The therapeutic potential was demonstrated via cell-death assays": this is not accurate.

3. Many English mistakes, including in the abstract. The grammar and spelling could be improved.

4. In 1D, they proved by IP the Anagrelide-dependent interaction of PDE3A and SLFN12. The same authors attempted that unsuccessfully in An et al, 2019, why is it now working? Is it properly explained in the Methods?

Reviewer #2 (Remarks to the Author):

In this manuscript, the authors report the cryo-EM structures of PDE3A-SLFN12 complex in the presence of Anagrelide or DNMDP, both work as molecular glues that promote the interaction between PDE3A-SLFN12, thereby inducing cell death. Based on the structural information, the author tested several synthesized compounds modified from Anagrelide, and found a novel compound with improved efficiency in promoting PDE3A-SLFN12 interaction. The major conceptual advances are (1) the structural basis for PDE3A-SLFN12 interaction, (2) mechanism of compound Anagrelide as a molecular glue, (3) Novel compounds with improved efficiency in inducing cell death via PDE3A-SLFN12 interaction. In all, I am enthusiastic about this study, although it looks that the manuscript has been prepared in a haste. In the meantime, some important issues require further investigation to make the study more complete.

Specific comments

1. The first section of the Results part is not novel. It has been reported in An et al. Am J Cancer Res 2019 9(9):1905-21 (This paper is in the reference list of this paper). Better put this part in the Supplementary information and make a proper citation in the Introduction part.
2. Page 2, paragraph 3, the authors wrote "Anagrelide induces the complexation of PDE3A and SLFN12 and prevents the SLFN12 degradation, and therefore increases the levels in living cells." If this was in a published article, please make a proper citation. Otherwise, the authors need to provide biochemical evidence for this statement. For example, test the binding affinity between PDE3A and SLFN12 with and without Presence of Anagrelide.
3. The authors reported the structure of PDE3A-SLFN12 complexed with DNMDP. However, DNMDP does not appear in the introduction at all, nor is this structure sufficiently describe or discussed. Please add necessary information for this structure, or remove it from the manuscript.
4. In the Discussion part, the authors mentioned that SLFN12 has a much narrower valley than SLFN13, and may "recognize distinct RNA substrates". I am curious about what substrates does SLFN12 have, and whether the biochemical property of SLFN12, as a potential DNA/RNA processing enzyme or DNA/RNA-binding protein, is essential for its cellular function, i.e. inducing cell death.

Minor points:

1. This manuscript contains no page number or line number, making it difficult to digest.
2. Page 3, paragraph 2, the whole paragraph is composed of a extremely dense sentence. Please rewrite this part to improve readability.
3. Page 4, first line, the SLFN13 structure reported in Yang et al, 2018 was from a truncated version that did not contain the SWADL domain (C-terminal as used in this manuscript, Figure 2D). Thus, to say "the density is missing" is not accurate. It is the "structural information" that is missing. Please also introduce the domain organization of the SLFN family proteins in the Introduction part.
4. This manuscript contains many typos and strange expressions in the context. Please carefully proofread the manuscript and make improvement in English writing.

Reviewer #3 (Remarks to the Author):

Dear authors,

with great interest, I read your manuscript entitled "Complex Structure of PDE3A–SLFN12 and Structure-based Molecular Glue Design for Apoptosis of Tumor Cells", submitted to Nature Communications. The manuscript reports on the importance of joint action of the phosphodiesterase 3A (PDE3A), the SFLN12 protein, and two novel PDE3A inhibitors, Anagrelide and DNMDP, for induced death of cancer cells. In particular, authors provide high-resolution x-ray models of catalytic domain of PDE3A incubated with both inhibitors in complex with the SLFN12 protein, derived from cryo-EM maps, and show how the inhibitors stabilize hydrophobic interactions of the catalytic domain of PDE3A and SLFN12. In addition, the team has selected 22 molecules as replacement candidates for anagrelide, and estimated their ability to induce cancer cell death. An anagrelide analogue with p-tolyl substitution at position 7 was identified as the most potent cell death inducer, and structural modeling suggested that this analogue indeed makes additional hydrophobic interaction with SLFN12.

I believe that the research is solid and I particularly like the screening of the anagrelide analogs. Unfortunately, the manuscript needs to be re-submitted because the initial submission lacks the EMDB validation reports of the cryo-EM maps, which are essential for the paper. The manuscript is also quite difficult to read and would definitely profit from professional touch rectifying both grammatical and stylistic errors.

In addition, I have come across several serious problems:

1) Interestingly, your expressed complex contained PDE3A in full length, but the high-resolution cryoEM map could be only obtained from its catalytic domain. The manuscript doesn't provide any evidence that the complex on the cryoEM grids really had PDE3 in full length. SEC-MALS or similar analyses are missing, cryoEM micrographs in supplementary data too small and class averages are masked. Please, rectify this problem. Please make sure that the readers know from the start that you worked with full-length proteins.

2) On the same note, I am missing the information on which part of the PDE3A could actually be modelled, i.e. the range of amino acids, in both the main text and figures.

3) Results, section 2, paragraph 4, and Fig. 2c: potential RNA binding region. "Two positively-charged patches responsible for RNA binding are located at the entrance of the valley (residues K35, K38, K39, R46 and R213, R217, K219)." Please reformulate the sentence because you haven't proposed evidence that this indeed is the RNA binding site, even if it looks promising.

4) Results, section 3, paragraph 1, sentence 1: Please indicate clearly that the aa669-11xx were INCLUDED, not excluded.

5) Results, section 3, paragraph 1, sentence 2: I disagree. Both truncations include the catalytic site (H756,H836,D837,D950), so they should behave identically in presence of Anagrelide if one considers this catalytic site. The change should be due to aa 670-679 or 1103-1141. Please clarify.

6) Discussion, paragraph2: Please clarify why you once use PDE3A and PDE3B another time.

7) Discussion, par. 4: "Importantly, we found that the Anagrelide and DNMDP likely shared the same binding pocket of cyclic nucleotides in PDE3A, based on the sequence conservation analysis and comparison with the structure of its isoform, PDE3B (Scapin et al., 2004). Therefore, the occupation by Anagrelide or DNMDP of PDE3A would preclude the binding of cyclic nucleotides, thus inhibiting its hydrolysis activity." However, in introduction, you state that de Waal et al found that "DNMDP can induce cell death by mediating the interaction of PDE3A and SLFN12, but without depending on the hydrolysis function of PDE3A". Could you clarify this point?

8) Figure 2c: please label the structure, highlight N-terminus, C-terminus, catalytic domain, compound binding pocket, etc, just like for SLFN12.

9) Figure 2e: ribbon representation of PDE3 with interacting residues would be much better.

10) Figure 3a: CD-5 should be CD-2

11) Figure 3d: I don't understand the scheme, not even with the help of the description.

12) Figure 3e: what are 1, 2, and 3 assigned to?

13) Figure 4b,c: again, ribbon representation of PDE3 with interacting residues would be much better than the surface, which would indicate the additional hydrophobic interaction more clearly.

14) Fig. 2a suggests that there are more interactions between PDE3 and SLFN12 than the short helix, as you claim in abstract. Is that true? Can you provide more details?

Minor points:

1) Please rectify grammar errors throughout the manuscript.

2) Please, bear in mind that the readers of multi-disciplinary journals don't need to possess deep biological knowledge. I would suggest to avoid using short designations like HeLa (SLFN12-/-)-SLFN12(K213R)-HA-3xFlag cells without thorough prior explanation or a list of abbreviations. Or a thorough description in the methods section. The same is true for the section heading: "4. Structure-guided optimization of Anagrelide led to A6 with significant increased activity." What is A6? We don't know it at this point of reading.

3) Examples of insufficient stylistics:

Introduction: 2nd sentence: "It contains three isoforms..." -> e.g. "It appears in three isoforms..."
par. 2: " The elevated SLFN12 binds to ribosome, thereby blocking the continuous protein translation, include Bcl-2 and Mcl-1, whose decrease triggers apoptosis cell death(Li et al., 2019)." - What does the ", include Bcl-2 and Mcl-1," mean here?

Results, section 2: "These hydrophobic interactions illustrate how these two molecules function in connecting PDE3A and SLFN12 as the molecule glue: molecule with one side at the pocket of PDE3A tightly anchored helix552-558 of SLFN12 by hydrophobic interaction." Which molecule? at which side?

Discussion, 1st paragraph: Here, we have known multiple ... -> Here, we have known THAT multiple ...

These small molecules binding PDE3A allows PDE3A to recruit cell death executor SLFN12 by interacting with SLFN12's helix552-558. -> Should probably be more like: The binding of these small molecules allows PDE3A to recruit

Despite my criticism, I believe that it is possible to improve your manuscript and include the missing EMDB cryoEM reports for re-submission.

With best regards!

Response to Referees letter

Re: NCOMMS-21-11653

“Complex Structure of PDE3A–SLFN12 and Structure-based Design for Potent Apoptosis Inducer of Tumor Cells”.

We appreciate all the referees for their instructive suggestions and comments. The referees’ comments on our original manuscript have been serving as a guide for us during our revision efforts. Listed below is the narrative summary of the changes made in the revised manuscript and the point-by-point responses to the referees’ comments were presented in the following pages.

Narrative summary:

1. We kept the arrangement of the original figures and refined the details of each figure to make the presentation more focused and precise. Specifically,
 - a) For **fig.1**, as reviewer#2 suggested, we removed the screening data and displaced it with the MK apoptosis-inducing data and immunoblotting data of the complex.
 - b) For **fig.2, C** was remade with the labeling of the subdomains for better illustrating the structural folding of PDE3A’s catalytic domain, as suggested by reviewer#3. And we add a new figure, **G**, to include the additional structure of nauclefine-induced PDE3A-SLFN12 complex to highlight the detailed structural information of how the apoptosis-inducing anagrelide, DNMDP, and nauclefine fit into the enzymatic pocket of PDE3A while use their hydrophobic moiety to provide the additional interactions with SLFN12 at the PDE3A-SLFN12 interface. Such information made it clear how these small molecules function as the molecular “Glue” to bring the otherwise non-interactive two proteins together. **A, B, D, E, F** are the same as the previous ones.
 - c) For **fig.3**, according to the question of reviewer#3, we redid the truncation experiments and replaced them with a new figure 3A and 3E in the revised manuscript, which is consistent with the previous conclusion. Figure 3E in the original manuscript was moved to the supplementary information.
 - d) For **fig.4**, as reviewer #3 suggested, we used ribbon representation of PDE3 with interacting residues. We rearranged **Figure 4** and presented the journey of structure-guided optimization in detail. The flowchart of our structure-based design strategy and the logic of compound development were also provided.
2. We added one figure (**fig. 5**) of the tumor xenograft growth inhibition data of anagrelide and its more efficacious analogs designed based on the PDE3A-SLFN12 complex structures was confirmed.
3. Since the nauclefine-induced PDE3A-SLFN12 complex was presented in the revised manuscript, we added the author, Shuanhu Gao, who synthesized the nauclefine, to the author list.

4. Additionally,

- 1) the full EMDB validation reports, map, and coordinate files were provided.
- 2) A zipped folder named 'Source Data' was provided.
- 3) And finally, all the new data and raw data associated with the paper have been deposited in a suggested repository, figshare.com with file#:10.6084/m9.figshare.14721444.
- 4) We have re-formulated the manuscript to fit the editorial requirement of Nature Communication.
- 5) We have done extensive re-writing and editing of the original manuscript, including the language polishing, grammar and spelling errors correction.

Our Response to the specific *Reviewers' Comments*:

Reviewer #1 (Remarks to the Author):

Chen et al report cryo-EM maps of Anagrelide (and DNMDP), as a “molecular glue” that enhances the interaction between SLFN12 and PDE3A. Structure-based design of analogs led to the development of a 20-fold more potent compound (A6). Anagrelide is a drug used for the treatment of thrombocytosis, and this study further supports the previously suggested mechanism of action of this drug.

Overall, this reviewer is not convinced about the novelty of this work. It is not clear how this work advances the field. Importantly, the major point number 1 mentioned below brings an important concern that needs to be clarified.

Our response: We thank this reviewer for this comments. We now have done extensive experiments and included new data to support the novelty of our work. We hope this reviewer will be convinced that our revised manuscript has been significantly improved in both novelty, and completeness over the original manuscript

MAJOR POINTS

1. The authors claim at the very beginning of the “Results” section: “Our screening efforts using an FDA-approved compound library against PDE3A showed that Anagrelide induces cell death in a dose-dependent manner.” As it is written, it implies that this screen was done now, as part of this paper, but no data is supporting it said screening was performed in a previous paper by the same authors that it is actually referenced in other sections: An et al. 2019. If there was not a new screening, the content and novelty of the paper are significantly lower than if this would have been the case.

Our response: The important findings reported in this manuscript are the previous un-reported high-resolution structures of PDE3A-SLFN12 complex with three small molecules that activate this pathway and how we used the detailed structural information to design more potent anagrelide analogs. We thus removed the section mentioning the previous screening effort that identified anagrelide as the apoptosis inducers that work through this pathway (referenced instead) and used new Figure 1 to first demonstrate that anagrelide induced the interaction of PDE3A and SLFN12 the cell death effect of anagrelide on megakaryocytes (MKs) apoptosis and the apoptosis of MK cells could be blocked by co-treatment of trequinsin, a PDE3A inhibitor.

2. The authors use the “molecular glue” concept a lot, including in the abstract. However, there is not a single article/review referenced neither in the introduction nor in the discussion. A proper introduction about molecular glues needs to be done.

Our Response: We appreciate this comment and indeed realized that word “molecular glue” was probably used in our original manuscript too liberally. In our revised manuscript, we provided content in the introduction and only introduced the concept in the context of the PDE3A-SLFN12 complexes that are “glued” together with the apoptosis-inducing small molecule anagrelide, DNMDP, and nauclefine. The related description in the revised Introduction is shown below:

“Molecular glues, small molecule compounds that bring otherwise non-interactive proteins to proximity for protein-protein interactions, have shown promising potential as therapeutical agents, especially for targeting the so-called “undruggable” proteins such as transcription factors and splicing factors. In general, molecular glues influence the activity or fate of their assembled proteins complex, either enhances the gain-of-function protein-protein interaction or induced the target protein degradation by bringing the cellular protein degradation machinery to act on the targeted protein.”

3. Why the authors decided to focus on Anagrelide (and also a bit in DNMDP) over Nauclefine is not explained.

Our response: We focused on anagrelide for the reason that it is a clinically-used drug with its pharmacokinetics, pharmacodynamics, and safety profiles well documented over the other two compounds. With this said, the other two compounds also provided valuable reference points for how these structurally diverse small molecules work in activating the PDE3A-SLFN12 pathway. In our revised manuscript, we did include the nauclefine-induced PDE3A-SLFN12 structure as well.

4. The authors claim in the introduction: “R. An. et al. (2019) showed that a PDE3A inhibitor (Anagrelide) induces interaction between PDE3A and SLFN12 and activates death signaling (An et al., 2019)”.

In the cited paper, it is instead stated:

“... Although we were unable to co-immunoprecipitated PDE3A and SLFN12 after ANA treatment, we did find that both PDE3A and SLFN12 were required...”

This point towards my comment raised just above, why did they focus on Anagrelide?

Our response: We appreciate this reviewer’s insightful comment on this point. An et al., 2019 was reporting the negative data that as we know now, may well be due to technical reasons. In this manuscript, by using the apoptosis induction defective K213R mutant, we were able to isolate enough nauclefine-induced PDE3A-SLFN12 complex to determine its cryo-EM structure. We included such structural information in our revised manuscript. The reason we focus on anagrelide is mentioned above.

5. These two claims are too speculative. If the authors want to include these statements, they should include data to show it:

- Speculation about steric effect preventing SLFN12 ubiquitin-mediated degradation

- In the discussion: the occupation by Anagrelide/DNMP of PDE3A would preclude the binding of cyclic nucleotides, thus inhibiting its hydrolysis activity.

Our response: We agree with this comment and have removed the first speculation in the revised manuscript.

As for the second point, it meant to state the fact that anagrelide and DNMDP shared the same binding pocket of cyclic nucleotides (the substrate of PDE3A), and these two molecules are well-known PDE3A inhibitors [*Nat Chem Biol.* 2016; 12(2): 102–108.]. Now we also find nauclefine also shares the same binding mode as the two other molecules. As we reported in Ai et al., 2020, nauclefine was unable to inhibit the cyclic nucleotide hydrolysis if enough cyclic nucleotide substrates were present. We also added sentences in the Discussion to clarify this point, as is shown below:

“Importantly, based on the sequence analysis and structure comparison with its isoform, PDE3B, we found that anagrelide, DNMDP, and nauclefine shared the same binding pocket of cyclic nucleotides in PDE3A. Therefore, the occupation by anagrelide, DNMDP, and nauclefine of the enzymatic pocket of PDE3A would in theory

preclude the binding of cyclic nucleotides, thus inhibiting its hydrolysis activity. However, our previous study on nauclefine indicated that such a molecule could also be readily competed out by the cyclic nucleotide substrates of PDE3A, making it appear that nauclefine was unable to inhibit the PDE3A enzyme activity when sufficient cAMP/cGMP was present in the reaction.”

6. To prove the molecular glue mechanism, it would be nice to show drug-induced proximity between PDE3A and SLFN12 using additional orthogonal strategies (split luciferase, alphaLISA, tethered drug analogs, biotin ligase, FRET...)

Our response: We appreciate these constructive suggestions. However, we do believe that the structure showing PDE3A-SLFN12 interaction requires additional binding surface from these small molecules provided the most direct evidence of molecular glue. The additional SAR of anagrelide analogs with more potent apoptosis inducer by increased the “glueness” of anagrelide further validated this concept.

MINOR POINTS

1. Figure 4D-G: IC₅₀s should be reported in the figure/legend/text.

Our response: We agree with the reviewer’s comments and the IC₅₀s have now been reported in the revised manuscript, and we have remade these figures (now Figure 4B-F in the revised manuscript). The revised Figure 4 as is shown below:

2. "The therapeutic potential was demonstrated via cell-death assays": this is not accurate.

Our response: We have modified this part by adding “*generated a compound that showed a much higher potency in inducing apoptosis in cultured cells and tumor growth inhibition in tumor xenografts*” in addition to the cell death assays in the revised manuscript.

3. Many English mistakes, including in the abstract. The grammar and spelling could be improved.

Our response: We have extensively re-written the text of the manuscript including the abstract to correct the grammar and spelling errors in the original manuscript.

4. In 1D, they proved by IP the Anagrelide-dependent interaction of PDE3A and SLFN12. The same authors attempted that unsuccessfully in An et al, 2019, why is it now working? Is it properly explained in the Methods?

Our response: As stated above, the unsuccessful IP of the PDE3A-SLFN12 complex in An *et al.* (2019) was due to technical issues. By using the K213R mutant, we are now able to IP the complex and resolved the cryo-EM structure that is reported in our revised manuscript.

Reviewer #2 (Remarks to the Author):

In this manuscript, the authors report the cryo-EM structures of PDE3A-SLFN12 complex in the presence of Anagrelide or DNMDP, both work as molecular glues that promote the interaction between PDE3A-SLFN12, thereby inducing cell death. Base on the structural information, the author tested several synthesized compounds modified from Anagrelide, and found a novel compound with improved efficiency in promoting PDE3A-SLFN12 interaction. The major conceptual advances are (1) the structural basis for PDE3A-SLFN12 interaction, (2) mechanism of compound Anagrelide as a molecular glue, (3) Novel compounds with improved efficiency in inducing cell death via PDE3A-SLFN12 interaction. In all, I am enthusiastic about this study, although it looks that the manuscript has been prepared in a haste. In the meantime, some important issues require further investigation to make the study more complete.

Specific comments

1. The first section of the Results part is not novel. It has been reported in An et al. Am J Cancer Res 2019 9(9):1905-21 (This paper is in the reference list of this paper). Better put this part in the Supplementary information and make a proper citation in the Introduction part.

Our response: We have removed the screening section to supplementary information and replaced the first part of the results with “Anagrelide-induced cell death in MK cells”, which explained how this drug is used to treat thrombocytosis.

2. Page 2, paragraph 3, the authors wrote “Anagrelide induces the complexation of PDE3A and SLFN12 and prevents the SLFN12 degradation, and therefore increases the levels in living cells.” If this was in a published article, please make a proper citation. Otherwise, the authors need to provide biochemical evidence for this statement. For example, test the binding affinity between PDE3A and SLFN12 with and without Presence of Anagrelide.

Our response: We have removed the statement and replaced it in the revised manuscript with the data showing anagrelide-dependent interaction of PDE3A and SLFN12.

3. The authors reported the structure of PDE3A-SLFN12 complexed with DNMDP. However, DNMDP does not appear in the introduction at all, nor is this structure sufficiently describe or discussed. Please add necessary information for this structure, or remove it from the manuscript.

Our response: We have now introduced DNMDP in the introduction section and the DNMDP-induced PDE3A-SLFN12 complex structure was also analyzed and discussed in the revised manuscript. The related description is shown below:

“In 2016, de Waal and their colleagues found that a synthetic chemical 6-(4-(diethylamino)-3-nitrophenyl)-5-methyl-4,5-dihydropyridazin-3(2H)-one, DNMDP, can induce cell death mediated by PDE3A and SLFN12 ”

“A structurally diverse group of chemicals, including 17- β -estradiol (E2), anagrelide, nauclefine, and DNMDP, all induce apoptosis by forming complexes with phosphodiesterase 3A (PDE3A) and Schlafen 12 protein (SLFN12).”

“DNMDP, can induce cell death mediated by PDE3A and SLFN12.”

“More recently, two more reports showed that two different PDE3A inhibitors, anagrelide and nauclefine, also induce apoptosis through this PDE3A and SLFN12 pathway and the apoptotic activity of DNMDP, E2, nauclefine and anagrelide can all be competitively inhibited by other PDE3A inhibitors, like cilostazol.”

“Using Flag- and HA-tagged SLFN12 to pull down PDE3A-SLFN12 complexes in the presence of anagrelide, nauclefine, or DNMDP, we obtained high-resolution cryo-EM structures of those protein complexes.”

4. In the Discussion part, the authors mentioned that SLFN12 has a much narrower valley than SLFN13, and may “recognize distinct RNA substrates”. I am curious about what substrates does SLFN12 have, and whether the biochemical property of SLFN12, as a potential DNA/RNA processing enzyme or DNA/RNA-binding protein, is essential for its cellular function, i.e. inducing cell death.

Our response: Thanks for raising this very interesting question. We have now removed this part in the revised manuscript because it is too speculative.

Minor points:

1. This manuscript contains no page number or line number, making it difficult to digest.

Our response: Page and line numbers have been added to the revised manuscript.

2. Page 3, paragraph 2, the whole paragraph is composed of a(n) extremely dense sentence. Please rewrite this part to improve readability.

Our response: We have revised this and rewritten the text as suggested.

3. Page 4, first line, the SLFN13 structure reported in Yang et al, 2018 was from a truncated version that did not contain the SWADL domain (C-terminal as used in this manuscript, Figure 2D). Thus, to say “the density is missing” is not accurate. It is the “structural information” that is missing. Please also introduce the domain organization of the SLFN family proteins in the Introduction part.

Our response: Thank you for bringing this to our attention. We corrected this error and provided the SLFN family domain organization information in the revised manuscript, as is shown below:

“Note that the structural information of the C terminal region is absent in previously published structures of SLFN family proteins.”

“The signature domain of human SLFN proteins is an AAA ATPase-like domain at their N-terminal regions with a highly conserved “SWADL” (Ser-Trp-Ala-Asp-Leu) domain positioned adjacent to the AAA domain. SLFN11, SLFN13, and SLFN14 have been shown to directly bind RNA. Human SLFN14 mutations underlie thrombocytopenia with excessive bleeding and platelet secretion defects but the molecular mechanism for such a defect is not known.”

4. This manuscript contains many typos and strange expressions in the context. Please carefully proofread the manuscript and improve in English writing.

Our response: We have done extensive re-writing and editing of the original manuscript.

Reviewer #3 (Remarks to the Author):

Dear authors,

with great interest, I read your manuscript entitled "Complex Structure of PDE3A–SLFN12 and Structure-based Molecular Glue Design for Apoptosis of Tumor Cells", submitted to Nature Communications. The manuscript reports on the importance of joint action of the phosphodiesterase 3A (PDE3A), the SFLN12 protein, and two novel PDE3A inhibitors, Anagrelide and DNMDP, for induced death of cancer cells. In particular, authors provide high-resolution x-ray models of catalytic domain of PDE3A incubated with both inhibitors in complex with the SLFN12 protein, derived from cryo-EM maps, and show how the inhibitors stabilize hydrophobic interactions of the catalytic domain of PDE3A and SLFN12. In addition, the team has selected 22 molecules as replacement candidates for anagrelide, and estimated their ability to induce cancer cell death. An anagrelide analogue with p-tolyl substitution at position 7 was identified as the most potent cell death inducer, and structural modeling suggested that this analogue indeed makes additional hydrophobic interaction with SLFN12.

I believe that the research is solid and I particularly like the screening of the anagrelide analogs. Unfortunately, the manuscript needs to be re-submitted because the initial submission lacks the EMDB validation reports of the cryo-EM maps, which are essential for the paper. The manuscript is also quite difficult to read and would definitely profit from professional touch rectifying both grammatical and stylistic errors.

Our Response: We have updated the validation reports including the EMDB validation results and did an extensive re-writing and editing of the original manuscript.

In addition, I have come across several serious problems:

1) Interestingly, your expressed complex contained PDE3A in full length, but the high-resolution cryoEM map could be only obtained from its catalytic domain. The manuscript doesn't provide any evidence that the complex on the cryoEM grids really had PDE3 in full length. SEC-MALS or similar analyses are missing, cryoEM micrographs in supplementary data too small and class averages are masked. Please, rectify this problem. Please make sure that the readers know from the start that you worked with full-length proteins.

Our response: PDE3A was in full length before loading onto EM grids, as evidenced by the gel below (Figure R1 and Figure S1C in revised supplementary information). We have added the full-length description of SLFN12 and PDE3A in the Results section, as is shown below:

“The purified complexes, as stained by Coomassie brilliant blue, showed protein bands correlating with that of full-length PDE3A and SLFN12 (Figure S1C), ...”

We enlarged the mask and performed the class-average analysis again, but still missed any extra density (Figure R2), so we believed that the N-terminal region of PDE3A was very flexible and averaged out during cryo-EM data processing. We have included a sentence to discuss this particular point in our revised manuscript, as is shown below:

“Notably, PDE3A in this study was from the endogenous origin and appeared as its full length in our purification (Figure S1C). Thus, the absence of N terminal density of PDE3A in our cryo-EM structures was likely due to the flexibility of the N terminus, which was averaged out during data processing.”

Figure R1. SDS-PAGE gel showing that PDE3A was purified as ~110 kDa, indicative of its full length (1141 amino acids).

Figure R2. 2D class-average results with an enlarged mask.

2) On the same note, I am missing the information on which part of the PDE3A could actually be modeled, i.e. the range of amino acids, in both the main text and figures.

Our response: We added the PDE3A's amino acids K669-Q1102 in the cryo-EM structure, as is shown below:

“...the PDE3A-SLFN12 complexes induced by all three compounds exhibited the same butterfly-like shape, appearing as a hetero-dimer of PDE3A and SLFN12 homo-dimer (Figure 2A and Figure S3), containing full-length SLFN12 and the C terminal region of PDE3A (i.e., the catalytic domain, K669-Q1102).”

3) Results, section 2, paragraph 4, and Fig. 2c: potential RNA binding region. "Two positively-charged patches responsible for RNA binding are located at the entrance of the valley (residues K35, K38, K39, R46 and R213, R217, K219)." Please reformulate the sentence because you haven't proposed evidence that this indeed is the RNA binding site, even if it looks promising.

Our response: We have now reformulated the sentence by removing "responsible for RNA binding", to reflect the speculative nature of such a statement.

4) Results, section 3, paragraph 1, sentence 1: Please indicate clearly that the aa669-11xx were **INCLUDED**, not excluded.

Our response: We have corrected this in the revised manuscript, as is shown below:

"PDE3A truncation variants, including residues 613-1108 (CD-1), 669-1108 (CD-2), 608-1141 (CD-3), 661-1141 (CD-4) and 679-1141 (CD-5) fused with a myc tag at its C-terminus, ..."

5) Results, section 3, paragraph 1, sentence 2: I disagree. Both truncations include the catalytic site (H756, H836, D837, D950), so they should behave identically in presence of Anagrelide if one considers this catalytic site. The change should be due to aa 670-679 or 1103-1141. Please clarify.

Our response: We examined the activities of five PDE3A truncations, including residues 613-1108, 669-1108, 608-1141, 661-1141, and 679-1141; we now present the full data in the revised manuscript. Four truncation variants, including residues 613-1108, 669-1108, 608-1141, and 661-1141 could rescue anagrelide-induced cell death, except for residues 679-1141. The data support that the change of cellular viability should be due to aa 669-679, but not 1108-1141. We have replaced the figure (the revised Figure 3A), as is shown below:

6) Discussion, paragraph2: Please clarify why you once use PDE3A and PDE3B another time.

Our response: The instance of "PDE3B" in the 2nd paragraph of the Discussion should be "PDE3A"; we have corrected this in the revised manuscript.

7) Discussion, par. 4: "Importantly, we found that the Anagrelide and DNMDP likely shared the same binding pocket of cyclic nucleotides in PDE3A, based on the sequence conservation analysis and comparison with the structure of its isoform, PDE3B (Scapin et al., 2004). Therefore, the occupation by Anagrelide or DNMDP of PDE3A would preclude the binding of cyclic nucleotides, thus inhibiting its hydrolysis activity." However, in

introduction, you state that de Waal et al found that "DNMDP can induce cell death by mediating the interaction of PDE3A and SLFN12, but without depending on the hydrolysis function of PDE3A". Could you clarify this point?

Our response: Based on the sequence conservation analysis and comparison with the structure of its isoform, PDE3B, anagrelide, nauclefine, and DNMDP likely share the same binding pocket of cyclic nucleotides in PDE3A. However, Li *et al.* (2019) study has shown that the enzymatic activity of PDE3A is not essential for E2-induced apoptosis. Two mutations of PDE3A at amino acid residues known to be critical for its enzymatic activity, including serine 465 and cysteine 816 could rescue E2-induced apoptosis in HeLa (PDE3A^{-/-}) with the endogenous PDE3A was knocked out. This indicated that E2/anagrelide/DNMDP induces apoptosis dependent on the enzymatic domain of PDE3A but did not require the hydrolysis function of PDE3A.

8) Figure 2c: please label the structure, highlight N-terminus, C-terminus, catalytic domain, compound binding pocket, etc, just like for SLFN12.

Our response: We have labeled the structure as suggested in the revised manuscript, as is shown below:

"..., which is composed of 16 α -helices divided into three subdomains: the N-terminus, M (middle), and C-terminal subdomains (Figure 2C). The catalytic site at the junction of these three subdomains was visible, comprising the H756, H836, D837, and D950 residues that chelate two metal cations. H752 at the N-terminal subdomain is known to function as a proton donor during cGMP/cAMP hydrolysis and this residue was also positioned nearby the catalytic site in our PDE3A model. The substrate-binding pocket is located at the C-terminal subdomain that anchors anagrelide, DNMDP, and nauclefine mainly through hydrogen-bonding interactions (Figure 2E-G)."

Accordingly, we also remade Figure 2C, as is shown below:

9) Figure 2e: ribbon representation of PDE3 with interacting residues would be much better.

Our response: We appreciate this suggestion and added a ribbon representation of PDE3A in the revised Figure 2E-G.

10) Figure 3a: CD-5 should be CD-2

Our response: We have corrected this in the revised manuscript. Thanks for pointing out this typo.

11) Figure 3d: I don't understand the scheme, not even with the help of the description.

Our response: We have corrected it in the revised manuscript. We remade the Figure 3D, as is shown below:

“The full-length *SLFN12* and truncation variants including residues 1-550, 1-560, and 1-570 fused with a Flag-tag at its C-terminus, were transfected in HeLa (*SLFN12*^{-/-}) cells. After treatment with anagrelide for 12 hr, *SLFN12* was immunoprecipitated using anti-Flag resin. The immunocomplexes were analyzed by immunoblotting using antibodies as indicated”.

12) Figure 3e: what are 1, 2, and 3 assigned to?

Our response: #1 here refers to the *SLFN12* truncation variant comprising residues 1-550, #2 refers to the *SLFN12* truncation variant comprising residues 1-560, and #3 refers to the *SLFN12* truncation variant comprising residues 1-570. Full-length of *SLFN12* and these three truncation variants were expressed in HeLa (*SLFN12*^{-/-}) cells from which the endogenous *SLFN12* locus was knocked out; we monitored the cell death using methyl blue staining.

To make this part more clear, we engineered several HeLa (*SLFN12*^{-/-}) cell lines in which the full-length *SLFN12* and three truncation variants are expressed under the control of a doxycycline (Dox)-inducible promoter. We moved figure 3E in the original manuscript to the supplementary information (Figure S8) and replaced it with the cell death using the Dox-inducible cell line in which expressed full-length *SLFN12* and truncation variants. Figure 3E in the revised manuscript, as is shown below:

13) Figure 4b,c: again, ribbon representation of PDE3 with interacting residues would be much better than the surface, which would indicate the additional hydrophobic interaction more clearly.

Our response: The point is well taken, and we have remade these figures (now Figure 4B-F in the revised manuscript) together with their IC50s, using ribbon representation of PDE3 with interacting residues, as is shown below:

14) Fig. 2a suggests that there are more interactions between PDE3 and SLFN12 than the short helix, as you claim in abstract. Is that true? Can you provide more details?

Our response: In addition to the described hydrophobic interactions between PDE3A and the short helix in SFLN12, several hydrogen bonds in the interface (Lys911, Asn1014, Gln1094 of PDE3A and Gly559, Ser368, Arg376 of SLFN12) may also contribute to the binding of PDE3A and SFLN12. Such a description is now included in the revised manuscript and Figure S7, as is shown below:

“The hydrophobic interaction mediated by anagrelide plays a central role at the PDE3A-SLFN12 heterodimer interaction interface, although there are also three hydrogen bonds dispersed at distant sites (Figure S7).”

Minor points:

1) Please rectify grammar errors throughout the manuscript.

Our response: The revised manuscript has been extensively re-written and edited.

2) Please, bear in mind that the readers of multi-disciplinary journals don't need to possess deep biological knowledge. I would suggest to avoid using short designations like HeLa (SLFN12^{-/-})-SLFN12(K213R)-HA-3×Flag cells without thorough prior explanation or a list of abbreviations. Or a thorough description in the methods section. The same is true for the section heading: "4. Structure-guided optimization of Anagrelide led to A6 with significant increased activity." What is A6? We don't know it at this point of reading.

Our response: We have now added detailed information in the Result and Methods section describing the generation of this cell line, as is shown below:

"HeLa (SLFN12^{-/-}) cells in which their endogenous SLFN12 was knocked out and were expressed Flag and HA-tagged SLFN12 (K213R) variant"

We also modified the text and Figure 4 to better introduce "A6" as one of the anagrelide analogs designed based on the structural information of PDE3A-SLFN12 complexes.

3) Examples of insufficient stylistics:

Introduction: 2nd sentence: "It contains three isoforms..." -> e.g. "It appears in three isoforms..."
par. 2: "The elevated SLFN12 binds to ribosome, thereby blocking the continuous protein translation, include Bcl-2 and Mcl-1, whose decrease triggers apoptosis cell death (Li et al., 2019)." - What does the ", include Bcl-2 and Mcl-1," mean here?

Our response: Thanks for pointing out these issues. We have extensively re-written the text for better clarity, as is shown below:

"It appears three isoforms in cells, each with the distinctive N-terminal regulator domain resulted from the alternative usage of the start codon and a common C-terminal catalytic domain."

"The elevated SLFN12 binds to the ribosome and the signal recognition particles (SRPs), thereby blocking the protein translation on the endoplasmic reticulum, including the translation of anti-apoptotic proteins Bcl-2 and Mcl-1, whose decrease triggers apoptosis."

Results, section 2: "These hydrophobic interactions illustrate how these two molecules function in connecting PDE3A and SLFN12 as the molecule glue: molecule with one side at the pocket of PDE3A tightly anchored helix552-558 of SLFN12 by hydrophobic interaction." Which molecule? at which side?

Our response: The molecules were specified in this context as anagrelide, nauclefine, and DNMDP in the revised manuscript, and this sentence was also revised accordingly, as is shown below:

"The hydrophilic moieties of these molecules containing carbonyl and amine groups anchor inside the PDE3A enzymatic pocket, while their hydrophobic aryl rings or alkyl tails extend outside of the pocket.... anagrelide is stacked with the phenyl ring in the side chain of PDE3A's F1004. SLFN12's helix552-558 is positioned at the mouth of the PDE3A enzymatic pocket, and the side chains of its L554, I557 and I558 residues comprise a highly hydrophobic moiety for interactions with the hydrophobic phenyl ring of anagrelide, together with the L910, I951, I968, F972, L1000 and F1004 residues of PDE3A (Figure 2E). These hydrophobic interactions

between these three molecules (anagrelide, DNMDP, and nauclefine) and SLFN12 “glue” the PDE3A and SLFN12 together.”

Discussion, 1st paragraph: Here, we have known multiple ... -> Here, we have known THAT multiple ... These small molecules binding PDE3A allows PDE3A to recruit cell death executor SLFN12 by interacting with SLFN12's helix552-558. -> Should probably be more like: The binding of these small molecules allows PDE3A to recruit

Our response: Again, we appreciate all the suggestions to improve our original manuscript. All of these suggestions are now incorporated into the revised manuscript.

REVIEWER COMMENTS

Reviewer #2 (Remarks to the Author):

Chen et al has made great effort in addressing the concerns from the referees. The scientific level and English writing have been evidently improved in the revised manuscript. I am satisfied with the submission in its current form. There are only some minor points as below.

1. Supplementary Figures were not sequentially quoted. For example, Page 5, Line 147: Figure S4A comes earlier than Figure S3A. I am not sure whether this is a problem for Nature Communications.
2. Page 5, Line 142: The headline of Section 2 was "glued" to the previous paragraph, although I think the Production Editor will solve this kind of mistakes.
3. Page 8, Line 220: "binds" should be "bind". There are also some other small typos or missing punctuation marks.

Reviewer #3 (Remarks to the Author):

Dear authors,

many thanks for all efforts that you have undertaken in the creation of the revised manuscript, for answering all questions, and clarification of ambiguities. Overall, the manuscript has improved a lot in my opinion, and I am happy with its quality as well as with the quality of the cryoEM maps and models you have built.

I have only few further suggestions and points to discuss:

- 1) Experiments on the determination of the PDE3A domain responsible for cancer cell death:
3. PDE3A's catalytic region binds to SLFN12's C terminal region upon anagrelide treatment

First of all, thank you for your clarifications. I fully agree that it seems to be the patch of AA 669-679 that is responsible for the PDE3A sensitivity to anagrelide treatment. Which is really interesting since these are actually the first few AA that could be modeled in your maps, and which are actually located rather far away from the catalytic and enzymatic site in the resolved maps. This probably suggest some allosteric interaction of the N-terminal domain that is required for proper functioning of PDE3A.

Therefore, I find the text on lines 211-214 still unclear on this point. I think it would be good to include your rebuttal sentence "The data support that the change of cellular viability should be due to aa 669-679, but not 1108-1141." into it in some way, and mention/discuss the possible role of the N terminal domain.

- 2) The missing density for the N-terminal domain: I agree that it is likely disordered, AlphaFold could predict only very few structured patches.
- 3) Methods: I haven't found a reference to HEK293T cells and virus packaging in the main text and supplementary material. Please check if these do need to be mentioned in the manuscript.
- 4) Despite much improved English grammar and stylistics, I still found quite some small errors. I am attaching annotated pdf file, in which I indicated possible errors with red handwriting and unclear sentences with pink highlighting.

With best regards!

Reviewer #4 (Remarks to the Author):

The manuscript entitled "Complex Structure of PDE3A–SLFN12 and Structure-based Design for Potent Apoptosis Inducer of Tumor Cells" described the drug design of novel apoptosis inducer targeting for complex of PDE3A–SLFN12 based on anagrelide. Reviewer thought that this manuscript has novel and important knowledge of the structure-activity relationship of anagrelide as an apoptosis inducer. The author responds and corrects scientifically and accurately to the reviewer's comments. Therefore, I judge that this manuscript has the scientific level accepted by Nature communications, but I will ask two additional questions.

1) In the structure-activity relationship of the anagrelide derivatives, the Cl group at the 7-position has not only a hydrophobic interaction but also an electrostatic interaction. Therefore, the author should also evaluate the activity of the derivative substituted with the OH group, which is the bioisostere of the Cl group. The author should also investigate the correlation between the hydrophobic parameter logP of all derivatives and antitumor activity.

2) From the HPLC chromatogram of the synthesized anagrelide derivatives, it appears that the two compounds are mixed with a shoulder at the main peak. The author needs to explain the reason for this.

Response to Referees letter

Our Response to the specific *Reviewers' Comments*:

Reviewer #2 (Remarks to the Author):

Chen et al has made great effort in addressing the concerns from the referees. The scientific level and English writing have been evidently improved in the revised manuscript. I am satisfied with the submission in its current form. There are only some minor points as below.

Our response: We appreciate your compliments and thanks for your endorsement!

1. Supplementary Figures were not sequentially quoted. For example, Page 5, Line 147: Figure S4A comes earlier than Figure S3A. I am not sure whether this is a problem for Nature Communications.

Our response: Thank you very much for pointing out this issue and we have switched the order of Figure S3 and S4.

2. Page 5, Line 142: The headline of Section 2 was “glued” to the previous paragraph, although I think the Production Editor will solve this kind of mistakes.

Our response: Thank you for your suggestion. We have corrected the format problem in our revised manuscript.

3. Page 8, Line 220: “binds” should be “bind” . There are also some other small typos or missing punctuation marks.

Our response: Thanks for pointing out these typos. We have carefully checked the whole manuscript and corrected all the grammar and spelling errors. All the revisions were highlighted in yellow.

Reviewer #3 (Remarks to the Author):

Dear authors,

many thanks for all efforts that you have undertaken in the creation of the revised manuscript, for answering all questions, and clarification of ambiguities. Overall, the manuscript has improved a lot in my opinion, and I am happy with its quality as well as with the quality of the cryoEM maps and models you have built.

Our response: We appreciate your insightful comment and strong supports!

I have only few further suggestions and points to discuss:

1) Experiments on the determination of the PDE3A domain responsible for cancer cell death:

3. PDE3A' s catalytic region binds to SLFN12' s C terminal region upon anagrelide treatment

First of all, thank you for your clarifications. I fully agree that it seems to be the patch of AA 669-679 that is responsible for the PDE3A sensitivity to anagrelide treatment. Which is really interesting since these are actually the first few AA that could be modeled in your maps, and which are actually located rather far away from the catalytic and enzymatic site in the resolved maps. This probably suggest some allosteric interaction of the N-terminal domain that is required for proper functioning of PDE3A.

Therefore, I find the text on lines 211-214 still unclear on this point. I think it would be good to include your rebuttal sentence "The data support that the change of cellular viability should be due to aa 669-679, but not 1108-1141." into it in some way, and mention/discuss the possible role of the N terminal domain.

Our response: Thank you for your constructive suggestions! We have added the sentence in the results section and discussed the possible role of the N terminal domain in the revised manuscript.

The revised part was also copied here:

"indicating that the change of cellular viability should be due to aa 669-679, but not 1108-1141"

"Our truncation experiments revealed that the patch of aa 669-679 of PDE3A is responsible for the PDE3A sensitivity to these small molecules treatment (Figure 3A), which is actually located rather far away from the catalytic and enzymatic site, i.e. the molecule-binding site. This probably suggests some allosteric interactions of the N-terminal domain that are required for the proper functioning of PDE3A."

2) The missing density for the N-terminal domain: I agree that it is likely disordered, Alphafold could predict only very few structured patches.

Our response: Thank you for your comments, and we agree with your opinion on Alphafold's prediction results as well.

3) Methods: I haven't found a reference to HEK293T cells and virus packaging in the main text and supplementary material. Please check if these do need to be mentioned in the manuscript.

Our response: Thank you for your comments. We have added the related reference in the revised manuscript:

"We transfected HEK293T cells with PDE3A truncation variants, including residues 613-1108 (CD-1), 669-1108 (CD-2), 608-1141 (CD-3), 661-1141 (CD-4) and 679-1141 (CD-5) fused with a myc tag at its C-terminus. These variants were stably expressed in a HeLa (PDE3A^{-/-}) cell line"

"Transfections and virus packaging. HEK293T Cells were transfected with plasmids using Lipofectamine 3000 (Thermo Fisher) following the manufacturer's instructions. Genes inserted into lentiviral-based vectors were co-expressed with psPAX2 and pMD2.G in HEK293T cells. Eight hours after transfection, the media were changed to high-serum DMEM (20% FBS with 25mM HEPES). Another 40 hours later, the media were collected and centrifuged at 3,000 rpm for 10 min. The supernatant was filtered through a 0.22-mm membrane and used to infect cells with polybrene (5µg/ml). Positively infected cells expressed GFP and were sorted by FACS to establish cell lines stably expressing specific genes".

4) Despite much improved English grammar and stylistics, I still found quite some small errors. I am attaching annotated pdf file, in which I indicated possible errors with red handwriting and unclear sentences with pink highlighting.

With best regards!

Our response: We really appreciate your time and efforts! Your corrections are remarkable and very helpful! We have corrected the issues in the revised manuscript according to your suggestions and highlighted them in yellow.

Reviewer #4 (Remarks to the Author):

The manuscript entitled “Complex Structure of PDE3A – SLFN12 and Structure-based Design for Potent Apoptosis Inducer of Tumor Cells” described the drug design of novel apoptosis inducer targeting for complex of PDE3A – SLFN12 based on anagrelide. Reviewer thought that this manuscript has novel and important knowledge of the structure-activity relationship of anagrelide as an apoptosis inducer. The author responds and corrects scientifically and accurately to the reviewer's comments. Therefore, I judge that this manuscript has the scientific level accepted by Nature communications, but I will ask two additional questions.

1) In the structure-activity relationship of the anagrelide derivatives, the Cl group at the 7-position has not only a hydrophobic interaction but also an electrostatic interaction. Therefore, the author should also evaluate the activity of the derivative substituted with the OH group, which is the bioisostere of the Cl group. The author should also investigate the correlation between the hydrophobic parameter logP of all derivatives and antitumor activity.

Our response: Thanks a lot for pointing out this interesting aspect of the bioisostere replacement, which is very helpful for our further lead optimization to the drug candidate. We also calculated the hydrophobic parameter logP of all derivatives and the correlation with antitumor activity was provided in the revised Supplemental Information. We did not observe the linear correlation of the logP value with IC₅₀ of all the molecules. However, as we indicated in the text, the hydrophobic interaction is crucial for the binding site of the PDE3A/SLFN12 interface. We then analyzed the correlation of the logP value and IC₅₀ of the molecules with the similar interaction pattern at this interface and indeed observed the same tendency that the greater the value of LogP, the lower IC₅₀ value. We admit that the interaction situation in this area is much more complicated and there are certainly other parameters that might affect the binding, such as the steric interruption, $\pi - \pi$ -stacking interactions and so on, but likely, the hydrophobic interaction plays a key role for the gluing activity in the current study.

2) From the HPLC chromatogram of the synthesized anagrelide derivatives, it appears that the two compounds are mixed with a shoulder at the main peak. The author needs to explain the reason for this.

Our response: Thanks for raising this HPLC chromatogram issue. We have re-done all the HPLC analyses with the same sample on a different column and no shoulder peaks were observed under the same separation condition. In addition, we re-checked the NMR data of all samples and the purities were also confirmed. So the

shoulder peak in HPLC might be due to the low separation resolution on the worn column. All the new HPLC data were provided in the revised manuscript. Thanks again for your insightful and constructive suggestions.